# CNN-Based Land Cover Classification Combining Stratified Segmentation and Fusion of Point Cloud and Very High-Spatial Resolution Remote Sensing Image Data

**Keqi Zhou [1], Dongping Ming [1,2,]\* , Xianwei Lv [1], Ju Fang [1] and Min Wang [3]**

1 School of Information Engineering China University of Geosciences (Beijing), 29 Xueyuan Road, Haidian, Beijing 100083, China
2 Polytechnic Center for Natural Resources Big-Data, MNR of China, Beijing 100036, China
3 Key Laboratory of Virtual Geographic Environment, Nanjing Normal University, Ministry of Education, Nanjing 210023, China
\* Correspondence: mingdp@cugb.edu.cn; Tel.: +86-10-13520907831

**Abstract:** Traditional and convolutional neural network (CNN)-based geographic object-based image analysis (GeOBIA) land-cover classification methods prosper in remote sensing and generate numerous distinguished achievements. However, a bottleneck emerges and hinders further improvements in classification results, due to the insufficiency of information provided by very high-spatial resolution images (VHSRIs). To be specific, the phenomenon of different objects with similar spectrum and the lack of topographic information (heights) are natural drawbacks of VHSRIs. Thus, multisource data steps into people's sight and shows a promising future. Firstly, for data fusion, this paper proposed a standard normalized digital surface model (StdnDSM) method which was actually a digital elevation model derived from a digital terrain model (DTM) and digital surface model (DSM) to break through the bottleneck by fusing VHSRI and cloud points. It smoothed and improved the fusion of point cloud and VHSRIs and thus performed well in follow-up classification. The fusion data then were utilized to perform multiresolution segmentation (MRS) and worked as training data for the CNN. Moreover, the grey-level co-occurrence matrix (GLCM) was introduced for a stratified MRS. Secondly, for data processing, the stratified MRS was more efficient than unstratified MRS, and its outcome result was theoretically more rational and explainable than traditional global segmentation. Eventually, classes of segmented polygons were determined by majority voting. Compared to pixel-based and traditional object-based classification methods, majority voting strategy has stronger robustness and avoids misclassifications caused by minor misclassified centre points. Experimental analysis results suggested that the proposed method was promising for object-based classification.

**Keywords:** data fusion; LiDAR; very high-resolution image; GeOBIA; stratified multiresolution segmentation; CNN

## 1. Introduction

With the development in artificial intelligence (AI) and remote sensing (RS) sensors in recent years, images with very high resolution are demanding more efficient processing means. Therefore, it is becoming increasingly hard to ignore deep learning (DL) models in object classification for RS images [1]. Deep learning which originated from artificial neural network (ANN) is currently widely adopted in diverse domains [2]. Among all DL methods, convolutional neural network is an approved method and has been successfully applied in RS.

Object classification generally consists of land-cover classification and land-use classification [3,4]. For urban areas, objects in remote sensing images are usually classified into diverse land-use categories, while objects are classified as land-cover sorts for non-urban areas. However, in this paper, object classification referred to land-cover classification for the lack of social attribute data. Desirable object classification was based on solid segmentation results. Different segmentation algorithms abound, and these methods can be divided into diverse types from different perspectives [5,6]. According to the nature of segmentation unit, segmentation algorithms are classified as pixel-level and object-level (pixel-based and object-based) segmentation [7–10]. The term of geographic object-based image analysis (GeOBIA) was first proposed by Blaschke [11]. For different experiment purposes, corresponding segmentation algorithms are selected. A superpixel-based segmentation method has been proposed for classification of very high-spatial resolution images (VHSRIs) [12,13]. Despite obvious advantages compared to pixel-based segmentation algorithms, GeOBIA still requires improvements. To further improve segmentation results, the strategy of stratification was introduced. Zhou et al. and Xu et al. [14] adopted grey-level co-occurrence matrix (GLCM) for a pre-segmentation which roughly segments the image into several homogeneously alike regions before multiresolution segmentation (MRS), which brought new possibilities for fine-MRS. Additionally, an algorithm adopting the voting strategy, which further increases classification efficiency was proposed for convolutional neural network (CNN)-based GeOBIA [15–18].

Deep learning is a currently adopted technique in GeOBIA-based land cover classification. Deep learning was reinterpreted in 2006 and has been booming ever since then under the unremitting efforts by Hinton [19]. The very first application adopts and realizes DL is from Lecun [20] and has been compared to traditional machine learning methods, including random forest and support vector machine, ever since [21]. However, LeNet fails to recognize images as complex as VHSRIs and is inefficient for land-cover classification in RS. DL consists of supervised learning, semi-supervised and unsupervised learning [22]. Among all models, CNNs (Convolutional Neural Networks) are one of the most frequently adopted deep supervised learning models. Alexnet, which won the Large Scale Visual Recognition Challenge 2012 (ILSVRC2012), successfully adopted rectified linear units (ReLU) as activate function and proposed LRN [23]. The merit of LRN is that it renders Alexnet a strong generalization capacity which enables Alexnet to learn and extract features from complex images including VHSRIs. Meanwhile, the advancement in GPU also encouraged the revival of machine learning. CNN has been applied in numerous domains, especially in computer vision related fields. In the RS field, CNN has been proven as a reliable tool for extraction and classification [24]. Scott et al. [25] further improves land-cover classification accuracy by adopting a deep CNN model. Apart from Alexnet, other newly proposed CNNs including VGGNet (Visual Geometry Group Network), ResNet (Residual Neural Network) and FCN (Fully Convolutional Network) are also successfully applied in GeOBIA [26]. Moreover, VGG which believes that CNNs with deeper architectures generate higher classification accuracy has been applied in VHSRI classification [27]. Apart from CNNs models, traditional machine learning (ML) algorithms and other DL methods perform effectively as well. Hong et al. [28] and Lu et al. [29] introduced the richer convolutional feature (RCF) to road and building edge detection in VHSRIs and overwhelmed traditional methods. Patch-based (CNN) and FCN are two of the most used models currently [30,31]. FCN outputs a result of the exact same size of input through deconvolution (backward learning). However, the structure of FCN is tedious. Even though the combining of CNN and RS images has generated great outcomes, a significant type of information, i.e., height, is missing in classification. Therefore, the introduction of light detection and ranging (LiDAR) becomes inevitable. Point cloud are data generated by LiDAR and are usually utilized as supplementary data in geoscience related fields. Multisource data fusion is commonly seen in RS and its concrete application such as object classification [32]. Multisource data fusion has always been an indispensable topic in RS since the beginning. Data sensed by diverse sensors, such as point cloud (PC), synthetic aperture radar (SAR), points of interests (POI), social sensing data [33,34] and surveyed data, have been applied in RS for deeper analysis in the past two decades [35–37]. Different data reflects unique features for diverse

objects. POIs refer to objects that attract specific researchers for a certain purpose and is promising for RS image classification. POIs can usually be buildings, bus stops, railway stations, hospitals, etc. Compared to RS imagery, POIs contain social attributes [38]. However, the addition of POIs has its own drawbacks that the information reflected by POIs may not be correct or timely. Similarly, social sensing data also reflects social properties that traditional RS data hardly shows [39]. Surveyed data reflects geometric information and may serve well as auxiliary data. Point cloud are generated by LiDAR laser sensors and mainly reveals elevation of the study area [40–42]. PC can be studied separately and be the auxiliary data in RS imagery analysis as well. The combination of PC and RS images demonstrates that the adjunction of elevation information further improves classification results [43,44].

However, although research adopting GeOBIA and DL in VHSRIs classification abound, as well as combining point cloud fusion in remote sensing land cover classification, more efforts are still potentially required. Firstly, due to that the complexity of scale dependence in segmentation of GeOBIA, fine segmentation is the key to GeOBIA classification. Secondly, from the perspective of classification features, how to effectively enhance the feature difference between different object classes or significantly increase the information entropy is the main purpose of data fusion. As a result, the performance of point cloud fusion and the quality of image segmentation should be further improved.

Aiming at these requirements, this paper explored a CNN based land cover classification method combining stratified segmentation and fusion of point cloud and VHSRI data, in which a new fusion named standard normalized digital surface model (StdnDSM) of PC and VHSRIs was first proposed in this paper, then a stratification strategy combining GLCM and MRS was applied to segment the fused data, and finally a finely tuned CNN model was utilized to train samples and classify land-cover objects. Image entropy was introduced to evaluate image quality for StdnDSM. For CNN based GeOBIA classification, the region majority voting strategy was applied to accelerate the procedure and avoid extreme situations that former methods fail to solve. A scene in Helsinki was chosen as the study area and the corresponding data was collected for study.

The remaining parts of this paper were settled as follows. Relevant critical methods and algorithms are well described in Section 2. In Section 3, the proposed StdnDSM and corresponding fused data are compared with former fused and solo data. Meanwhile, the adoption of GLCM in MRS and utilization of PC in segmentation and classification were proved as superior. Pros and cons of the experiment are analysed and discussed in Section 4 and finally in Section 5 conclusions are drawn.

## 2. Materials and Methods

In this part, four parts are introduced in succession. Firstly, the novel proposed StdnDSM is minutely explained. Secondly, GLCM and MRS along with their relationship are fully demonstrated. Then, the CNN model and classification strategy adopted in experiments are illustrated separately. Eventually, accuracy assessment methods are presented. The workflow for the CNN based land cover classification method combining stratified segmentation and point cloud data fusion is demonstrated in Figure 1.

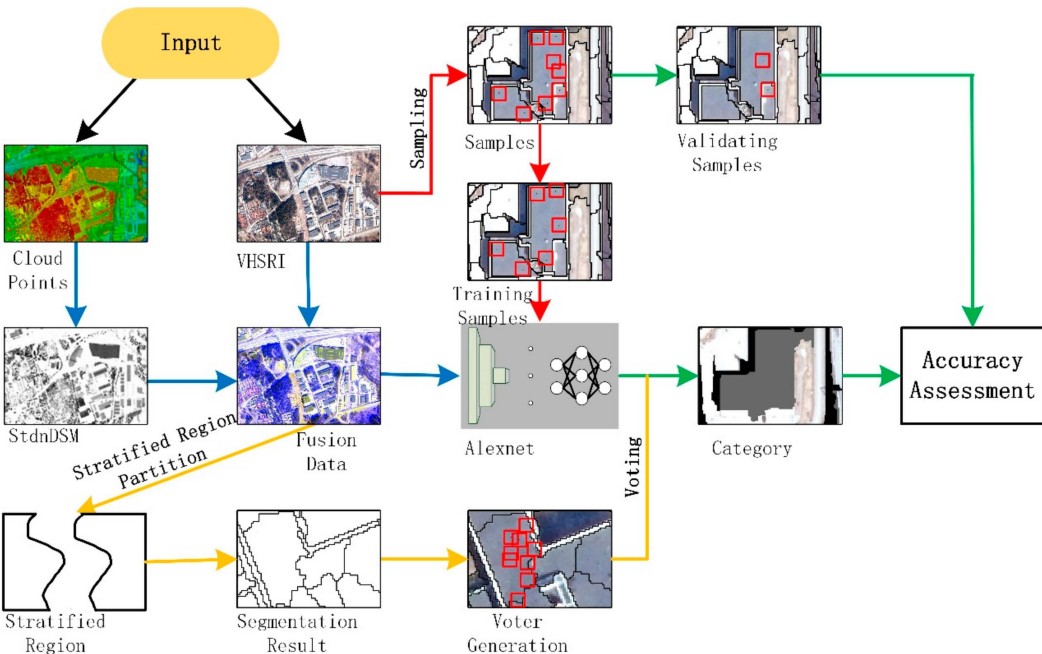

**Figure 1.** The workflow for the convolutional neural network (CNN)-based land cover classification method combining stratified segmentation and point cloud data fusion.

## 2.1. StdnDSM for Point Cloud Data Fusion

Before StdnDSM is described in detail, three terms, digital terrain model (DTM), digital surface model (DSM) and normalised DSM (nDSM; also known as digital height model, DHM), ought to be introduced. Moreover, all four digital models here are generated from PC. DTM directly reflects the elevation of the ground. DTM consists of a grid model, triangulated irregular network (TIN) model and contour model, among which grid DTM is most ideal for DL input. Equation (1) shows how DTM is obtained from PC by Kriging interpolation:

$$Z(x_0) = \sum_{i=1}^{n} \lambda_i Z(x_i) \tag{1}$$

where $Z(x_0)$ is the required elevation of a certain point and $Z(x_i)$ are the elevation of known points nearby. The $Z(x_0)$ is actually the weighted mean of $Z(x_i)$ and $\lambda_i$ is the weight of neighbouring point $x_i$. However, DTM is hardly suitable for data fusion with VHSRIs as segmentation and classification are object-based while DTM fails to emphasize objects. Similarly, DSM (digital surface model) is deficient for fusion or DL.

Thus, nDSM (normalized digital surface model) which is the outcome of subtracting DTM from DSM was proposed to be fused with VHSRIs. Equation (2) presents the generation of nDSM:

$$nDSM = DSM - DTM. \tag{2}$$

The advantage of nDSM, compared to DTM and DSM, is that nDSM only reflects relative heights. To put it another way, objects such as bare lands and waters read zero in digital number (DN) value while objects including vehicles, trees and buildings are distinctly above zero. Hence, differences among objects in elevation are reflected by nDSM. Yet, nDSM is scarcely perfect for fusion with VHSRIs as the D-values between nDSM and VHSRIs are commonly tremendous. Since nDSM is generated from PC whose accuracy in height reaches centimetre level (airborne-based LiDAR), its DN value floats within thousands. However, for VHSRIs or ordinary RS images, DN values are usually restricted to a certain range, e.g., 0–255 for 8 bit images, 0–65536 for 16 bit images. Therefore, there is an issue of

dimensional disunity between nDSM and DN of VHSRI. nDSM may overwhelm VHSRIs if the two are directly fused. Given such circumstances, considering the uniform of dimension, StdnDSM was proposed in this paper to fully and reasonably combine PC and VHSRIs. StdnDSM was obtained by standardly constraining the DN value of nDSM within the same range of DN of VHSRI. In Equation (3), StdnDSM is defined as follows:

$$\text{StdnDSM} = \delta \left[ \frac{DN - DN_{Min}}{DN_{Max} - DN_{Min}} \right] \tag{3}$$

where $\delta$ is the extent of RS image which is to be fused with StdnDSM. DN is the digital number of a certain pixel, $DN_{Max}$ is the maximum digital number around the image and $DN_{Min}$ is the minimum digital number of the image. In Figure 2, original PC data is illustrated along with StdnDSM. The eventual aim for proposing StdnDSM was to convert cloud point data which was non-suitable for DL into an ideal alternative raster image that mathematically surpassed nDSM with richer information.

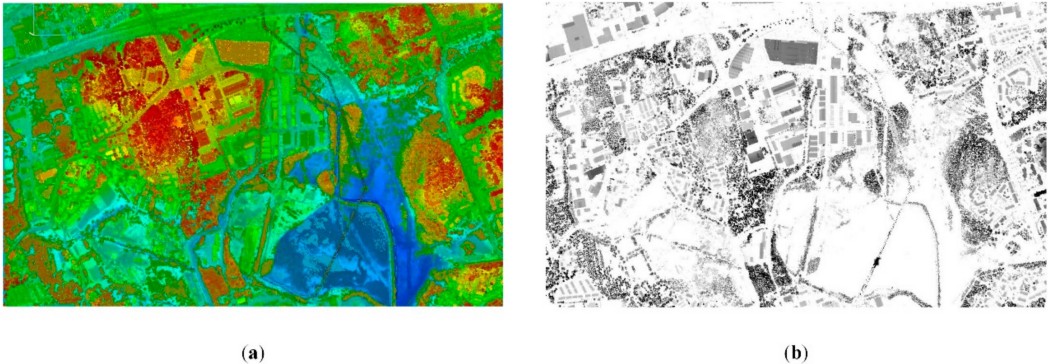

(a)                                                                                   (b)

**Figure 2.** The comparison of point cloud (**a**) and standard normalized digital surface model (StdnDSM) (**b**). The site was located in Helsinki, Finland.

From the comparison of DTM, DSM, nDSM and StdnDSM shown in Figure 3, it can be vividly seen that StdnDSM and nDSM beat the other two images visually, especially for urban green belts and buildings. Urban green belt and building areas in DTM and DSM visibly appeared to be blurrier compared to nDSM and StdnDSM. The visual effect of DTM was completely ambiguous and basically deficient for any object-level segmentation or classification. As for DSM, essential figures of buildings and woods were distinct while they were mixed in border areas. However, for nDSM and StdnDSM, practically all objects were legible. The reason for such an outcome was that nDSM and StdnDSM removed influences of ground surface and contained only objects heights. The cause behind the excellence of StdnDSM in data fusion was the absence of potential excessive dominance over VHSRIs. That is, in fusion data the influence of elevation reflected by StdnDSM (PC) and spectral information contained in VHSRIs were equally distributed.

As Figure 4 shows, the corresponding fusion data of DTM (a), DSM (b), nDSM (c) and StdnDSM (d) are compared. All fusion data were visually promising for classification while the fusion of StdnDSM and VHSRI remained as the optimal. It was apparent that the fusion data of StdnDSM and VHSRI was spectrally homogenous, which was the credit of confining elevation information consistent with spectral information in the DN value. To put it differently, StdnDSM surpassed DTM, DSM and nDSM in data fusion with RGB images for adding elevation into RS data.

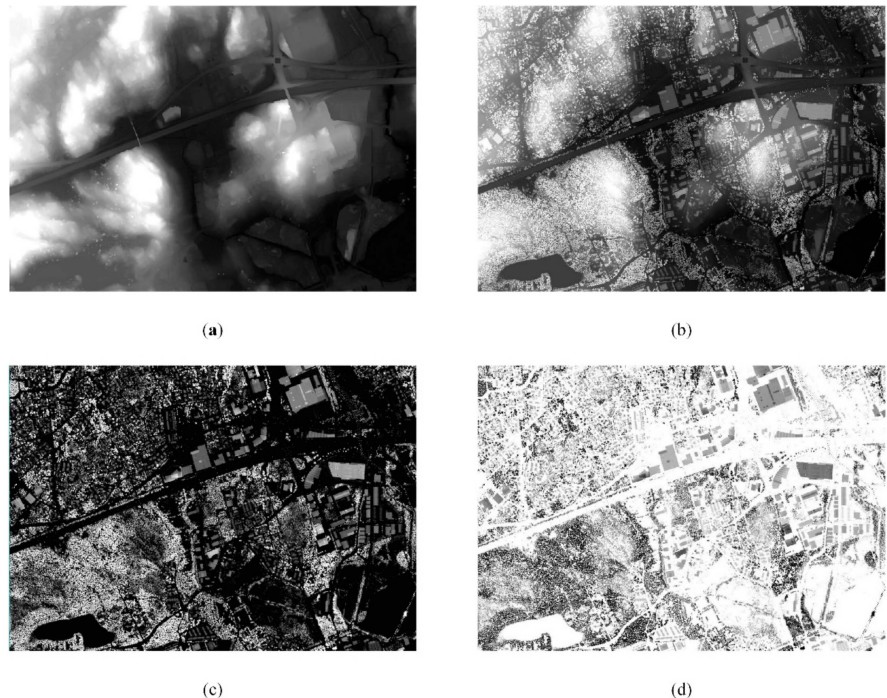

**Figure 3.** The comparison of (**a**) digital terrain model (DTM), (**b**) digital surface model (DSM), (**c**) normalised DSM (nDSM) and (**d**) StdnDSM. The site was located in Helsinki, Finland.

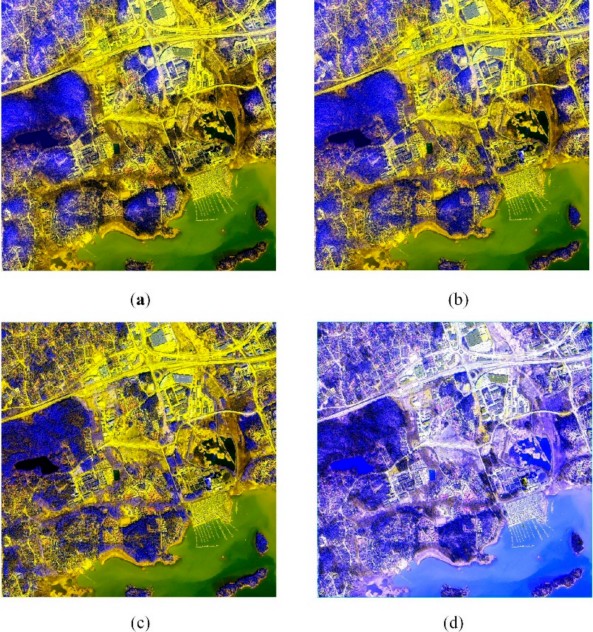

**Figure 4.** The comparison of fusion data. (**a**) DTM, (**b**) DSM, (**c**) nDSM and (**d**) StdnDSM. The site was located in Helsinki, Finland.

*2.2. Stratified Segmentation*

As for land-cover classification, objects of diverse sizes including vehicles, buildings and waters were distributed throughout the research area. A uniquely fixed parameter in MRS hardly performs perfectly for all those objects. Put differently, the best segmentation scale for buildings was not the same as water or other objects. Therefore, GLCM based region partition was involved before the fine scale segmentation to improve segmentation efficiency and then heighten classification accuracy. The co-occurrence matrix which was a function of the distance and angular relation between two

neighbouring pixels reflected the times a pixel and its specific neighbouring co-occurrence [45,46]. GLCM considers eight directions including 0°, 45°, 90°, 135°, 180°, 225°, 270° and 315° between a pair of neighbouring pixels in specific distance d [47]. In Figure 5, an exhibition of the GLCM working principle is presented. Figure 5a shows the eight viable processing directions. Figure 5b presents a demonstration of a 3 × 3 window as it moves along "1,0" (90°) direction and Figure 5c is the related GLCM calculation result.

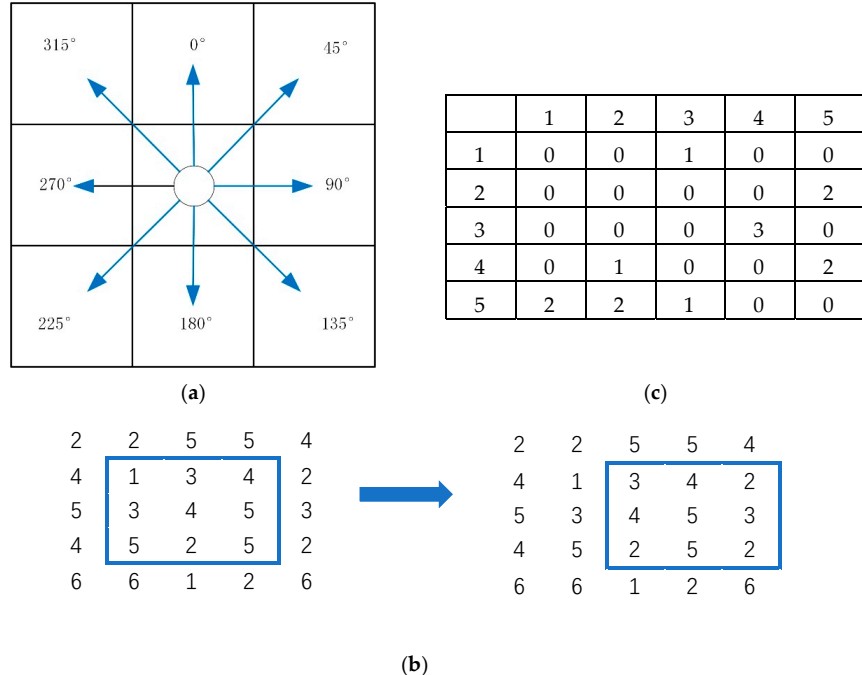

**Figure 5.** A demonstration of gray-level co-occurrence matrix (GLCM). (**a**) The eight processing directions of GCLM, (**b**) the processing window moves and (**c**) the counted co-occurrence matrix.

GLCM in application acted as texture filters based on the co-occurrence matrix. Those filters, namely mean, variance, homogeneity, contrast, dissimilarity, entropy, second moment, correlation, etc., denote respective texture features [48]. Twenty-three of twenty-eight filters, including contrast, second moment and correlation, were deemed as irrelevant in this experiment and shall not be introduced next. The remaining five filters, i.e., mean, variance, homogeneity, dissimilarity and entropy, are explained and illustrated by Equations (4)–(8). Equation (4) shows how the local mean value of the processing window is obtained.

$$\text{Mean} = \sum_{i}\sum_{j} P(i,j)/N_g{}^2 \tag{4}$$

where $P(i,j)$ denotes the value of $i$th row and $j$th column in grey-the level co-occurrence matrix while $N_g$ refers to the number of grey levels. The variation which shows the extent how pixels in GLCM change is presented as Equation (5).

$$\text{Var} = \sum_{i}\sum_{j}(i - Mean)^2 p(i,j) \tag{5}$$

where $p(i,j)$ is the $(i,j)$th entry in a normalized grey-tone spatial-dependence matrix and equals $P(i,j)/R$. $R$ is the number of neighbouring resolution cell pairs for generating the grey-tone

spatial-dependence matrix. In Equation (6), homogeneity which is calculated adopting the inverse difference moment algorithm is demonstrated:

$$\text{Hom} = \sum_i \sum_j \frac{1}{1 + (i-j)^2} p(i,j). \tag{6}$$

Filter homogeneity emphases pixels with co-occurrence relation in texture space. On the contrary, filter dissimilarity amplifies the unlikeness between neighbouring pixels in texture and is defined as Equation (7):

$$\text{Dis} = \sum_{n=1}^{N_g - 1} n \frac{\left\{ \sum_{i=1}^{N_g} \sum_{j=1}^{N_g} p\{i,j\} \right\}}{|i-j| = n}. \tag{7}$$

The absolute D-value of greyscale, $|i-j| = n$, is used here to acquire dissimilarity. The last filter adopted in this paper was entropy which reflects the redundancy of texture information and is explained as Equation (8):

$$\text{Ent} = -\sum_i \sum_j p(i,j) \log(p(i,j)). \tag{8}$$

The outcome of stratification strategy adds texture information in MRS, which improves segmentation in both result and efficiency. However, the final segmentation result was determined by stratification strategy and MRS. The reason for selecting MRS before classification was that MRS performs better in consistency and most importantly efficiency than other segmentation methods. For large research areas, the efficiency of segmentation greatly influences experimental process. The essence of MRS is fractal net evolution approach (FNEA) [49]. In a certain scale, objects diverse in size who seldom share the same finest segmentation resolution synchronously exist. Thus, MRS performs a pixel-level bottom-up local grow segmentation which strictly adheres to maximum homogeneity/minimum heterogeneity criterion to merge spectrally homogeneous pixels. However, not only spectral feature but also the shape is considered as an indispensable part in MRS. In Equation (9), the permissible heterogeneity upper bound *Hetero* which decides the area of segmentation objects is defined.

$$\text{Hetero} = \omega_{color} \times hete_{color} + \omega_{shape} \times hete_{shape} \tag{9}$$

where $\omega_{color}$ and $\omega_{shape}$ are respectively the weight of spectral and shape feature while $hetero_{color}$ and $hetero_{shape}$ denote heterogeneity of objects in spectra and shape. In Equations (10) and (11), $hetero_{color}$ and $hetero_{shape}$ are further introduced.

$$hetero_{color} = \sum_x \omega_x (n_a(hete_{mc} - hete_{ax}) + n_b(hete_{mc} - hete_{bx})) \tag{10}$$

where $x$ means a certain band in the image and $\omega_x$ refers to the weight of this band in all bands. $n_a$ and $n_b$ are the number of pixels in object a and object b who are to be merged. $hetero_{ax}$ and $hetero_{bx}$ are respectively the heterogeneity of object a and object b in band $x$ while $hetero_{mc}$ denotes the heterogeneity of potential merged object c.

$$hetero_{shape} = \frac{l}{\sqrt{n}} \tag{11}$$

where $l$ refers to the perimeter of the possible object c while n is the number of pixels in object c.

As is known, the similar objects always cluster in the same local region with similar sizes. Thus, dividing the VHSRI and finely extracting the object in local regions can improve the scale suitability and accuracy. By combining GCLM based region partition and MRS, this paper introduced a stratified MRS method. The stratified MRS method segmented the whole image into several large regions before independently segmenting each region at different scales.

### 2.3. Convolutional Neural Network and Alexnet

Convolutional neural network which was enlightened by visual perception mechanism of cats is an indispensable subfield of deep learning. Four fundamental ideas behind CNN [50] respectively are local connections, shared weights, pooling and utilization of numerous layers. Diverse CNN algorithms abound in tremendous domains, however there are three basic layers, namely convolutional layer, pooling layer and fully-connected layer, which remain unchanged. Convolutional layers extract local conjunctions from features, and then similar features are merged into one by pooling layers. Eventually, all newly obtained features are combined by fully-connected layers. The natural advantage of CNN is avoiding over-fitting compared to traditional DL algorithms. Moreover, images can be directly input into CNN, which prevents intricate data pre-processing such as data reconstruction. Whereas, due to the black-box mechanism, features extracted are unintelligible and inexplicable, which is the double-edged sword for CNN and brings unreliability and infinite possibilities simultaneously. It is worth noting that the input of images is confined to raster data, which was unsatisfactory for this paper owing to the introduction of LiDAR data.

Alexnet, the champion of ILSVRC2012, is known as a breakthrough in CNN, which was why it was selected in this experiment. Five convolutional layers, three pooling layers and two fully-connected layers are contained in Alexnet. Numerous filters are embodied in convolutional layers to extract features. Rectified linear units (ReLU) is adopted as the activation function to constrain the range of values. In Equation (12), the ReLU activation function is presented.

$$a_{i,j,k} = max\big(Inp_{i,j,k}, 0\big) \tag{12}$$

where $Inp_{i,j,k}$ is the function input at location $(i, j)$ on $k$th channel. An appropriate loss function is demanded for a specific experiment. For this paper, the softmax loss was chosen. Figure 6 depicts the architecture of Alexnet.

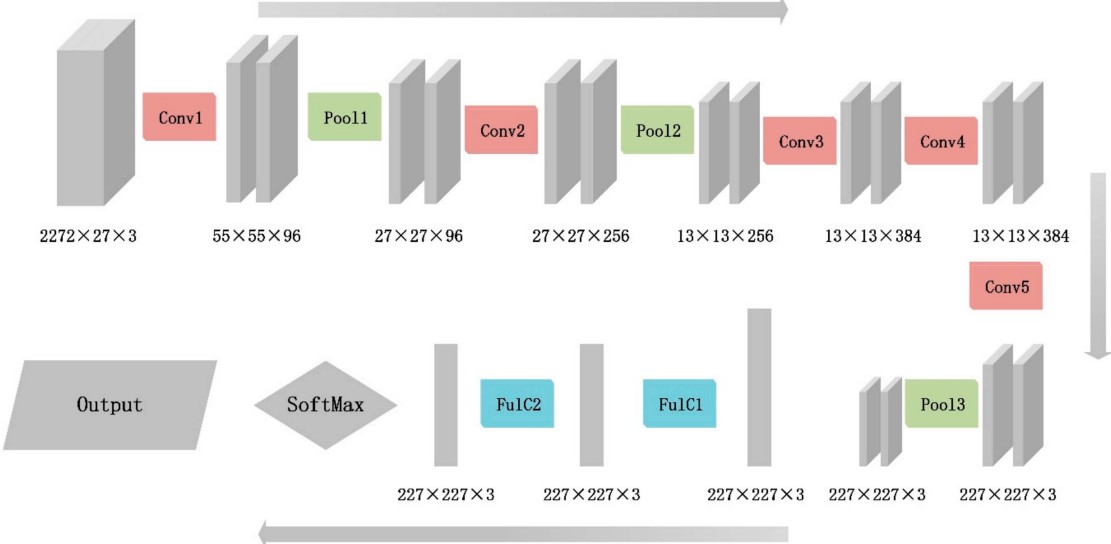

**Figure 6.** The architecture of Alexnet. Conv represents the convolutional layer; Pool denotes pooling layer; FulC is the fully-connected layer. Numbers underneath each image indicate corresponding size and dimension.

### 2.4. Region-Based Majority Voting

Object-based classification results are based on and considerably influenced by segmentation results. For object-wise CNN, casting CNN training outcomes into segmentation results is unavoidable. RMVCNN is a strategy connecting MRS results and CNN training results. Majority voting strategy

consists of two parts, i.e., voters generation and vote. Voters directly engage with CNN, while vote is more postprocessing.

The key thought of majority voting is that each segmented polygon shall contain odd points acting voters which are later labelled by CNN with an attribute. In other words, the final category of a segmented polygon is determined by labelled voters within. The ideal situation is that each polygon has odd voters, which prevents any possibility of a tie vote. Thus, voters generation generates odd points within each segmented polygon. On the basis of the sizes of objects, the number of voters shall accordingly vary. For example, objects, such as vehicles and ships, normally contain dozens of pixels and corresponding segmented polygons are of the small size. A single voter is sufficient for such objects. Whereas, larger objects including buildings require more voters to determine categories. Therefore, all segmented polygons are graded according to size before generating voter(s). To ensure the validity of voter(s), the centre point of polygon and even random points within are required. The number of even random points is decided by the grade of each polygon in size.

Vote refers to rendering segmentation results with voters' labelled attributes. Each polygon and its voter(s) shall be spatially joined before the vote. The procedure of vote is demonstrated in Figure 7.

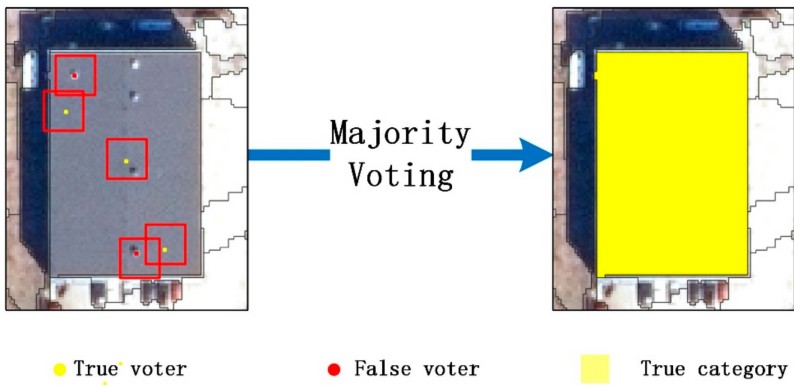

**Figure 7.** Demonstration of majority voting.

There are five voters within the shown polygon, two of which are labelled as road while the other three ones are building. In the light of vote principle, the category of this polygon is then set as building. The adoption of majority voting effectively decreases the chances of misclassification caused by segmentation results. Meanwhile, multiple voters are more persuasive than one centre point to determine the category of an object.

*2.5. Accuracy Assessment Methods*

The accuracy assessment of this classification experiment is based on two essential indexes, namely overall accuracy (OA) and kappa coefficient. These two indexes can be obtained from the confusion matrix which is illustrated in Table 1.

**Table 1.** The illustration of the confusion matrix.

|  |  | Actual Class | | |
| --- | --- | --- | --- | --- |
|  |  | **A** | **B** | **C** |
|  | A | $X$ | $Y_1$ | $Z_1$ |
| **Predicted Class** | B | $X_1$ | $Y$ | $Z_2$ |
|  | C | $X_2$ | $Y_2$ | $Z$ |

For Table 1, $x_{true}$ distributed on the diagonal is the number of accurately predicted object A, while $y_{true}$ and $z_{true}$ are respectively that of object B and object C. Unfilled blocks are mis-predicted numbers of each object. Thus, overall accuracy $P_o$ and kappa coefficient are acquired as Equations (13) and (14).

$$P_o = \frac{x_{true} + y_{true} + z_{true}}{n} \tag{13}$$

$$k = \frac{P_o - P_e}{1 - P_e} \tag{14}$$

where $n$ is the sum of objects and $P_e$ is the theoretical agreement rate which can be calculated as Equation (15).

$$P_e = \frac{\sum N_{act} \times N_{pre}}{n^2} \tag{15}$$

where $N_{act}$ is the actual number and $N_{pre}$ is the predicted number of each object. The kappa coefficient is confined to $[-1, 1]$ but is commonly above zero. Moreover, it consists of five levels, i.e., slight (0.0–0.2), fair (0.21–0.4), moderate (0.41–0.6), substantial (0.61–0.8) and almost perfect (0.81–1).

Apart from the accuracy assessment, information entropy (IE) was introduced to assess and compare nDSM and StdnDSM. Referenced from the second law of thermodynamics, the information entropy in Information Science was proposed by C. E. Shannon [51,52] to assess the amount of information contained in a certain information source. Information entropy is actually the numerical statement of information source and is related to the probability of random events which in raster image refers to single pixels. The definition of information entropy is represented in Equation (16):

$$H(X) = C \sum_{i=1}^{n} p(x_i) \log p(x_i). \tag{16}$$

## 2.6. Experiment Description

This section consists of two major parts, Section 3.1 Experiment Description and 3.2 Experiment Result and Comparison. In this section, the study area and data of this experiment, how sampling and stratified MRS were performed and CNN tuning and training are introduced in detail. The classification result of the experiment is shown and compared in Section 3.2. Furthermore, it should be noted that this experiment was performed on a machine learning machine equipped with Window10 OS and two NVIDIA TITAN X 12 Gb GPU. Alexnet was chosen as the fundamental CNN frame for its high performance in object classification and built on Tensorflow 1.7.0. Data processing were performed on professional software including QGIS (mapping and analysing), ArcGIS (mapping), ENVI (nDSM), LiDAR360 (DTM and DSM generating) and eCognition (segmentation).

### 2.6.1. Image Description and Data Fusion

The study area was a district in Helsinki, the capital city of Finland, which also is one of the biggest port city in Europe. The comprehensiveness of Helsinki ensured that the study area covered nearly all object categories. Meanwhile, VHSRIs and LiDAR data were provided by the National Land Survey of Finland (NLSF) and is demonstrated in Figure 8. The image size is 5995 × 5995.

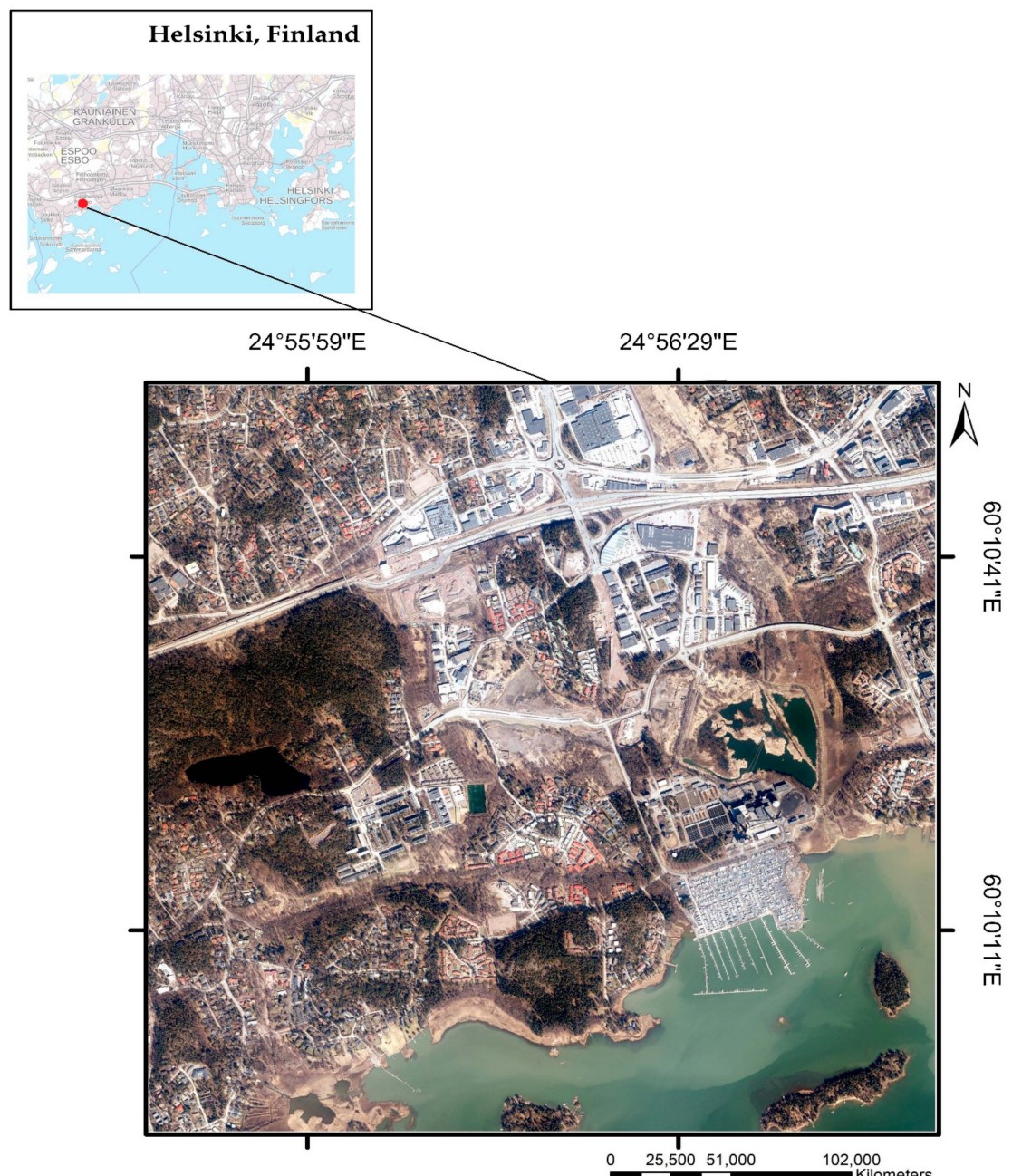

**Figure 8.** Experiment area: a scene in Helsinki.

It should be noted that the coordinate system of both VHSRI and point cloud data are ETRS-TM35FIN. The VHSRI was captured in August 2014 by an airborne sensor and uploaded to NLSF on 9 February 2017. As the image was sensed in summer, vegetations were spectrally easy to be distinguished and objects such as snowfield were avoided. The image resolution was 0.5 m/pixel. The point cloud data was produced on 22 August 2008 by airborne sensor Optech ALTM GEMINI and uploaded to NLSF on 16 December 2016. The elevation precision was 0.15 m while the average point density (resolution) was 43 pis/m$^2$. The inconsistency in time phase brought challenges in data fusion. However, such difficulties could be overcome thanks to the world's leading urbanization in Helsinki. In other words, although the imaging time of VHSRI and PC are eight years apart, the city appearance of study area barely varies. Therefore, data fusion in this experiment was feasible and worth trying. Nevertheless, the high resolution of both VHSRI and PC ensured the possibility of land cover wise classification.

As the input of Alexnet is confined to raster images, PC needed to be transformed and then fused with VHSRI. Cloud points data was first transformed into DTM and DSM to further generate nDSM. This operation was performed on LiDAR360 software. In the processing, the cell assignment type was set by inverse distance weighting (IDW) and nearest neighbour method was used as void fill algorithm. The proposed StdnDSM which resolves the unequal weight phenomenon of directly fused VHSRI and PC was acquired based on nDSM. Figure 9 compares the StdnDSM and nDSM shown by a single band image.

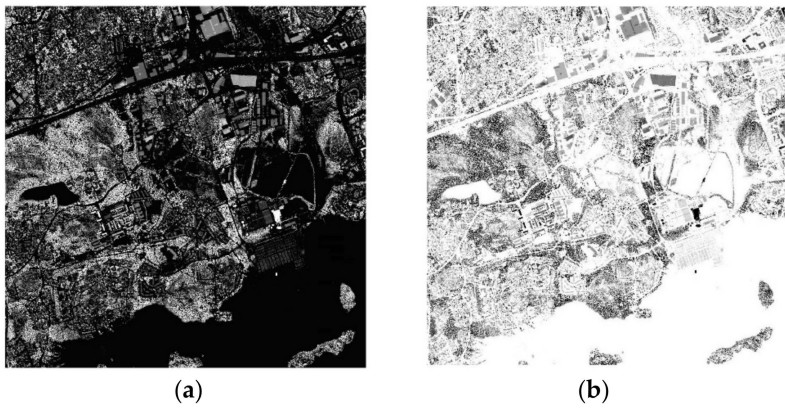

(**a**) 　　　　　　　　　　　　　　　 (**b**)

**Figure 9.** The comparison of nDSM (**a**) and StdnDSM (**b**).

The difference between those two was hardly visible but was reflected in the final results and information entropy assessment shown following. Assessments on StdnDSM, nDSM and corresponding fusion data were performed, whose outcome shown in Figure 10 suggested that StdnDSM contained greater amount of information compared to nDSM and StdnDSM fusion data beats nDSM fusion data. Moreover, it should be noted that higher IE does not necessarily indicate better performance in classification, but indeed proved the superiority of StdnDSM.

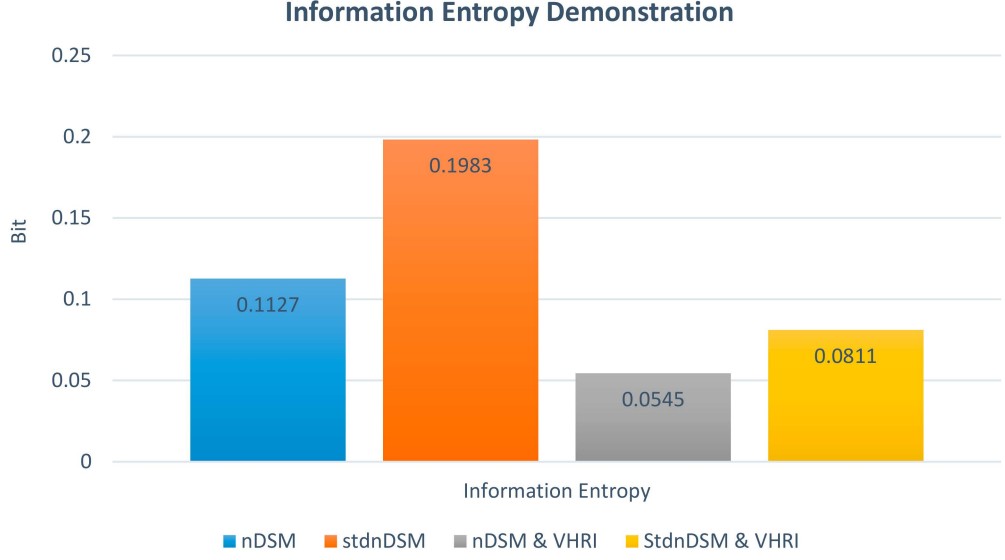

**Figure 10.** Information entropy demonstration of nDSM, StdnDSM, nDSM fusion and StdnDSM fusion.

Meanwhile, the advantage of StdnDSM was proven from the data fusion mechanism. Once the StdnDSM was generated, three bands were selected from band R, G, B, and StdnDSM for Alexnet supported only three channels. To choose the ideal band combination, principal components analysis (PCA) was adopted for its ability to select bands with the maximum amount of information. The result

of the PCA is shown in Table 2, according to which band R, band G and StdnDSM were selected for PC data fusion.

**Table 2.** The result of principal components analysis.

| | Correlation Matrix | | | |
|---|---|---|---|---|
| **Layer** | **R** | **G** | **B** | **nDSM** |
| R | 1 | 0.0794 | 0.17983 | 0.11541 |
| G | 0.0794 | 1 | 0.9668 | 0.93594 |
| B | 0.17983 | 0.9668 | 1 | 0.97563 |
| nDSM | 0.11541 | 0.93594 | 0.97563 | 1 |

After the optimum band combination was confirmed, data fusion was performed whose result is demonstrated in Figure 11 of which the RGB channels respectively correspond to red, green and StdnDSM. It should be emphasized that Figure 11 was the input to Alexnet

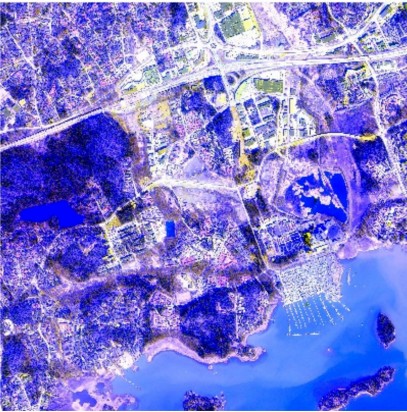

**Figure 11.** The training image of Alexnet.

### 2.6.2. Sampling for Training and Testing

Sampling is of great significance in semi-supervised classification as the classifier is not capable of rendering an attribute to a to-be classified object independently. In other words, sampling is the procedure where hidden attributes and learning objects are linked. Without proper training samples, CNN fails to distinguish different objects or is likely to be confused and then misclassify objects. After all, human–machine coupling is believed to be the future of AI. Apart from appropriate training samples, good validating samples are also demanded as the final OA is based on them. To ensure the reliability of OA, training samples and validating samples are required to abide by the same sampling criterion and strictly avoid repeated samples. If the same samples are adopted as training samples and validating samples simultaneously, the obtained OA will inflate compared to the genuine OA.

To meet above requirements, training samples and validating samples of this experiment were selected at the same time. To be specific, samples were distributed into training samples and validating samples after the sampling had been done. Therefore, this strategy ensured that there were no points that were concurrently training and validating points. According to the peculiarity of the study area, the categories of land-cover were classified into nine sorts, namely "building", "dock", "road", "tree", "water", "ship", "vehicle", "shadow" and "bare land". Each category of samples was selected individually so that different categories could be independently and purposefully fused for contrast experiments. The selected samples are demonstrated in Figure 12 and Table 3. As shown in Figure 12, red points represented training samples and consisted of 25,000 samples, while yellow points denoted validating samples and were of 3811 samples.

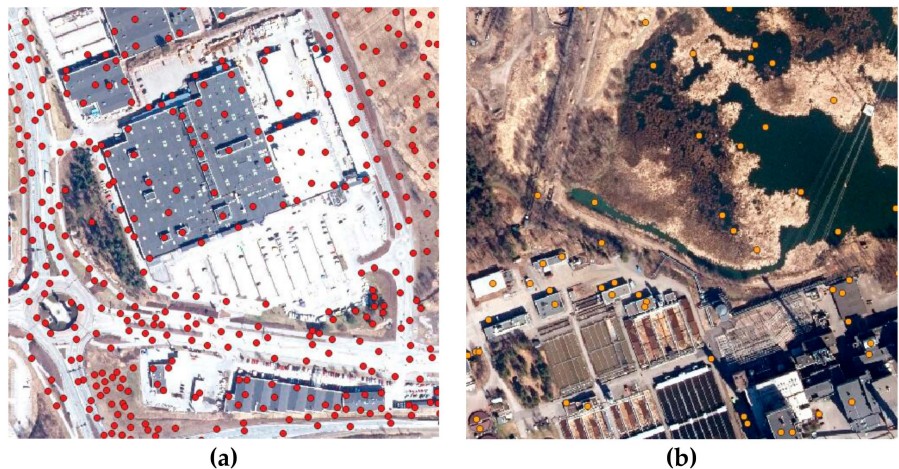

**Figure 12.** Demonstration of (**a**) training samples and (**b**) validating samples.

**Table 3.** Amount of each category in training samples.

| Category | Bare Land | Building | Dock | Road | Shadow | Ship | Tree | Vehicle | Water |
|---|---|---|---|---|---|---|---|---|---|
| Amount | 1594 | 6871 | 882 | 2659 | 2941 | 2344 | 6147 | 659 | 903 |

### 2.6.3. Stratified MRS

As mentioned above, the introduction of GLCM is an outcome of stratification strategy. As demonstrated in Figure 13, four homogenous regions respectively contained a type of dominating objects. The adoption of MRS tremendously decreases the number of segmented objects in segmentation results compared to traditional segmentation methods. However, there still remain the predicament that diverse objects share the same segmentation scale, which makes the segmentation result less justified theoretically. It is apparent that the best segmentation scale of vehicles and buildings should not be the same. Nine categories of land-cover objects interpenetrated in the study area. However, the distribution of objects was not completely random. Some similar objects tended to cluster while other objects were hardly possible to exist simultaneously. For example, vehicles could never be within water areas but were always close to roads. In other words, the first law of geography still applies well here. Thus, a region partition on coarse scale before MRS was necessary for a better segmentation result. In this experiment, the GLCM was performed to add texture information to the pre-segmentation and pre-segmented the study area into four large homogenous regions. As mentioned previously, five indexes, namely mean, variance, homogeneity, dissimilarity and entropy were chosen and then fused with band red, green, blue and StdnDSM for pre-segmentation. The reason for selecting those five filters was that they contained more information than other filters visually and statistically. By using GLCM, four homogenous regions respectively containing a type of dominating object were partitioned as demonstrated in Figure 13.

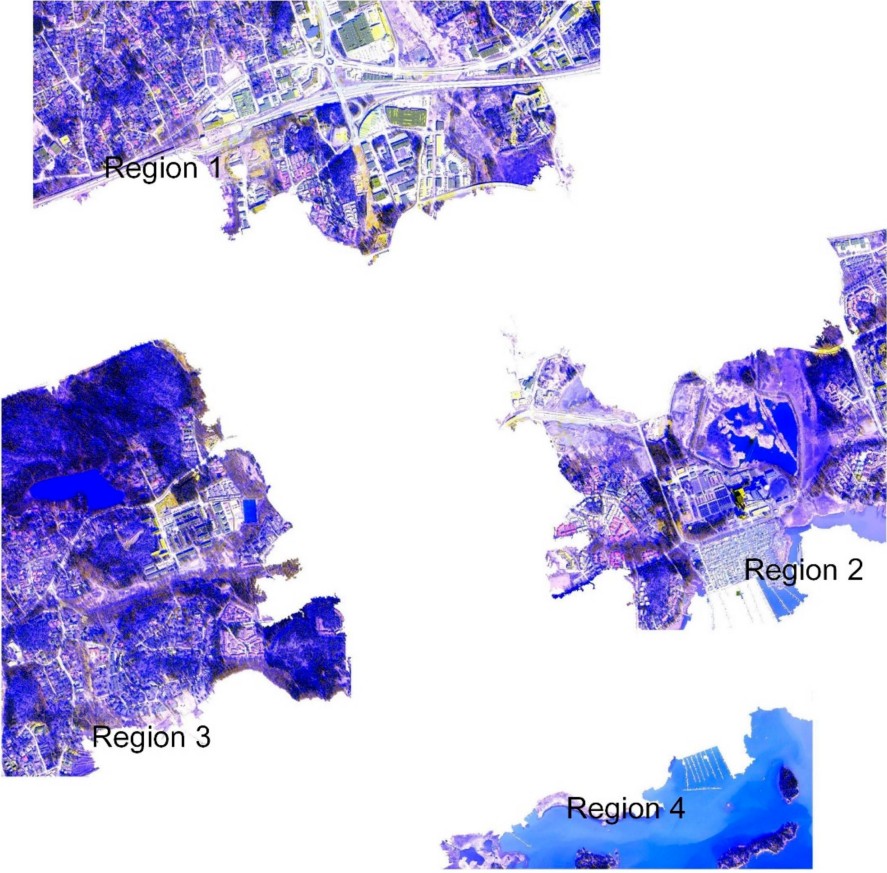

**Figure 13.** Large homogenous regions.

In the north-west region (Region 1), trees were the most distributed objects while water dominated the south-east region (Region 4). According to the sizes of dominating objects in each large homogenous region, different segmentation scales were chosen for the four regions. The best segmentation scales of all four regions are respectively shown in Table 4.

**Table 4.** The segmentation scales of each homogenous region.

| Parameters / Region | Region 1 | Region 2 | Region 3 | Region 4 |
|---|---|---|---|---|
| Scale Parameter | 20 | 20 | 20 | 20 |
| Shape | 0.7 | 0.3 | 0.5 | 0.4 |
| Compactness | 0.3 | 0.7 | 0.5 | 0.6 |
| Band Ratio | 1:1:1:7 | 1:1:1:7 | 1:1:1:7 | 1:1:1:1 |

The best segmentation scales of each large homogenous region were the result of trial and error. Adequate trials were performed to guarantee that the best scales adopted were reliable. However, it should be emphasised that the band ratio of Region 4 in MRS was set as 1:1:1:1 which was different from other areas. The reason behind such a decision was that the elevation of waters was less significant in segmentation. The greatest advantage of adopting GLCM for a pre-segmentation was making MRS more theoretically explainable and efficient. In the subsequent section "Classification Results and Overall Accuracy", a detailed comparison of adopting and non-adopting GLCM MRS will be presented in detail. Within each segmented object (polygon), several odd voters (voting points) were then generated for majority voting. The total amount of voters was 278,836. Thanks to the stratification strategy, the number of voters shrank by hundreds of thousands, which will be presented in Section 3.2.

### 2.6.4. CNN Training and Classification

As one of the most mature CNNs, the Alexnet was adopted in this experiment. There are several critical parameters that require detail stated. Above all, the learning rate controlling learning process was set as 0.01, as it is commonly adopted by tremendous experts in RS. Then, the number of epochs which refer to the procedure that all data travel through the workflow of CNN for one time was set as 160 as the training accuracies of experimented CNNs tended to be saturated after 150th epochs and more epochs would be simply pointless. Thirdly, the number of batches which evenly divided training samples into certain subsets as the amount of samples was too great to be directly put into CNN was set as 100. To avoid over-fitting, the dropout rate of Alextnet was set as 0.5. Eventually, as this paper considered scale effect in classification, experiments at more than one scale were performed. The scale of training was determined by the window size of CNN. To guarantee the comprehensiveness of this paper, four single scales and ten related scale combinations were independently performed. Four single scales were respectively 15, 25, 35 and 45. Ten scale combinations were 15–25, 15–35, 15–45, 25–35, 25–45, 35–45, 15–25–35, 15–25–45, 15–35–45 and 25–35–45. Results of each training scale will be demonstrated in Section 3.2.

Apart from the parameters discussed above, it should be also noted that the data ratio and training ratio were both set as 0.8. In other words, 20,000 out of 25,000 training samples were randomly input into CNN and 16,000 out of 20,000 training samples were directly taken as training data. The remaining 4000 training samples were used for CNN self-regulation. Once CNN training was done, a tuned network was obtained for further classification.

As explained previously, the fused data was first segmented into polygons, in which voters were generated for rendering attributes. Therefore, the classification of segmented objects was actually determined by the classification of voters. Once voters were input into tuned Alexnet, the network classified by rendering an attribute to each voter according to the features it learned. Then, voters with attributes were utilized to determine the attribute of each segmented and to-be classified objects. As each polygon contained multiple voters, the final attribute of it was democratic and reasonable.

Moreover, the comparative experiment where the B channel was nDSM was performed along with the experiment above. The corresponding results of comparative experiments are to be introduced in detail in Section 3.2 Classification Results and Accuracy.

## 3. Results

As together segmentation and classification for segmented objects composed the object-based CNN classification, comparisons will be demonstrated from three aspects namely, results, OA and the efficiency. Efficiency is about segmentation, while results and OA are related to classification.

### 3.1. Efficiency

Improvements in the efficiency of the introduced method were mainly reflected in segmentation. As the stratified MRS was a GEOBIA method, segmentation was of great significance in final classification results. Therefore, a stratified MRS method was proposed to enhance segmentation in both efficiency and fineness. The efficiency of the introduced method is reflected by statistics in Table 5 while the fineness is shown in Figure 14.

**Table 5.** Comparison of stratified and unstratified multiresolution segmentation methods.

| Segmentation Methods | Polygon Amount | Voters Amount |
|---|---|---|
| Stratified MRS of Fusion Data | 94,144 | 278,836 |
| Unstratified MRS of Fusion Data | 106,985 | 297,227 |
| Unstratified MRS of VHSRI | 202,364 | 340,978 |

As Table 5 shows, stratified MRS of fusion data generated least both polygons and voters among all three methods, which indicated the superiority of both stratification strategy and fusion data (added LiDAR data). The voters generating principle was strictly region oriented. To be specific, polygons sized from 0–50 $m^2$ contained only one voter (centre point), polygons sized from 50–100 $m^2$ contained three voters, polygons sized from 100–500 $m^2$ contained five voters and polygons that were large than 500 $m^2$ contained seven voters. Such a principle was believed as both efficient and rational. It was unnecessary to specify polygons sized from 100–500 $m^2$, as they were basically same or solo similar objects. Moreover, the comparison of unstratified MRS of fusion data and VHSRI greatly showed the advantage of point cloud which reflected elevation information. Polygons generated by directly segmenting fusion data was merely half the number of polygons obtained by segmenting VHSRI and the number of voters decreased by over 43 thousand. It should be noted that the parameters of these two unstratified MRS methods were exactly the same. With different voters generating principles, specific voter numbers of diverse methods may vary. However, the improvement in efficiency of introduced stratified MRS method was solid and nonnegligible.

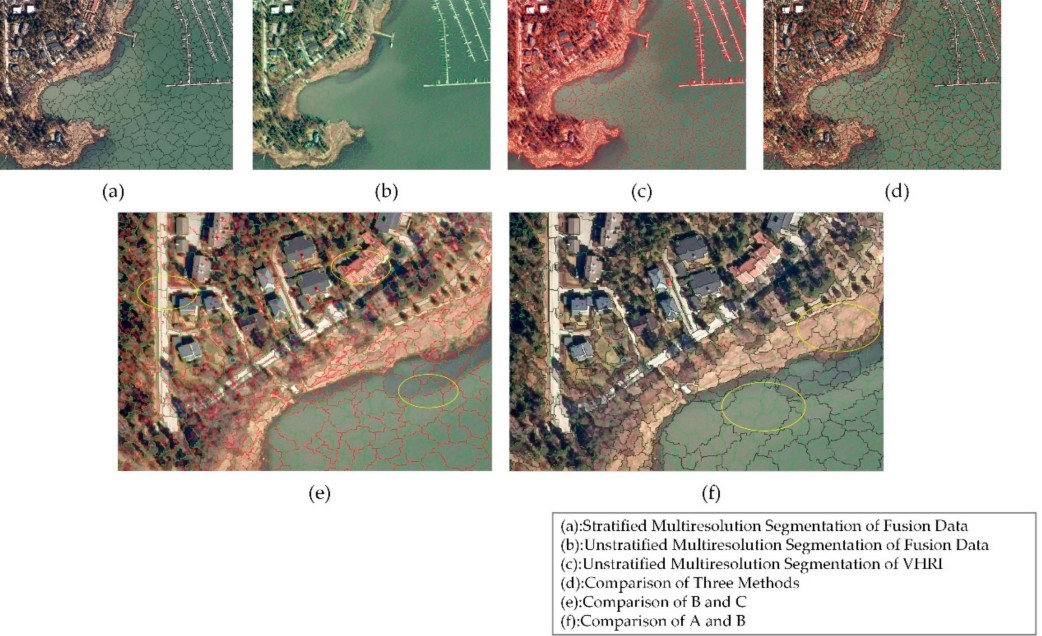

(a)                    (b)                    (c)                    (d)

(e)                                    (f)

(a):Stratified Multiresolution Segmentation of Fusion Data
(b):Unstratified Multiresolution Segmentation of Fusion Data
(c):Unstratified Multiresolution Segmentation of VHRI
(d):Comparison of Three Methods
(e):Comparison of B and C
(f):Comparison of A and B

**Figure 14.** The comparison of stratified and unstratified multiresolution segmentation (MRS) of fusion data.

Figure 14 visually demonstrates the superiorities of the stratified MRS and fusion data. Figure 14a shows the adopted segmentation method, that is stratified MRS of fusion data, whose segmentation result was the finest among those three methods, especially around the borders of large regions mentioned above. Demonstrated regions located at the border of the water region and the mixed region which consisted of almost all types of objects. It was apparent that for Figure 14b,c, segmentation results of unstratified MRS of fusion data, were not as good as the adopted method. To be specific, Figure 14e denoted that for objects of high homogeneity such as roads and waters, the addition of LiDAR surpassed single-source data in efficiency. That is, segmentation result of green lines was obviously less dense than that of red lines. Furthermore, Figure 14f showed that the adopted stratified MRS performed better than direct MRS, as the result of black lines was more efficient than that of green lines.

### 3.2. Classification Results and Accuracy

In this section, the final results of classification are to be shown and compared. Meanwhile, OAs of all classification results will be compared.

The classification results are shown and compared in Figures 15 and 16.

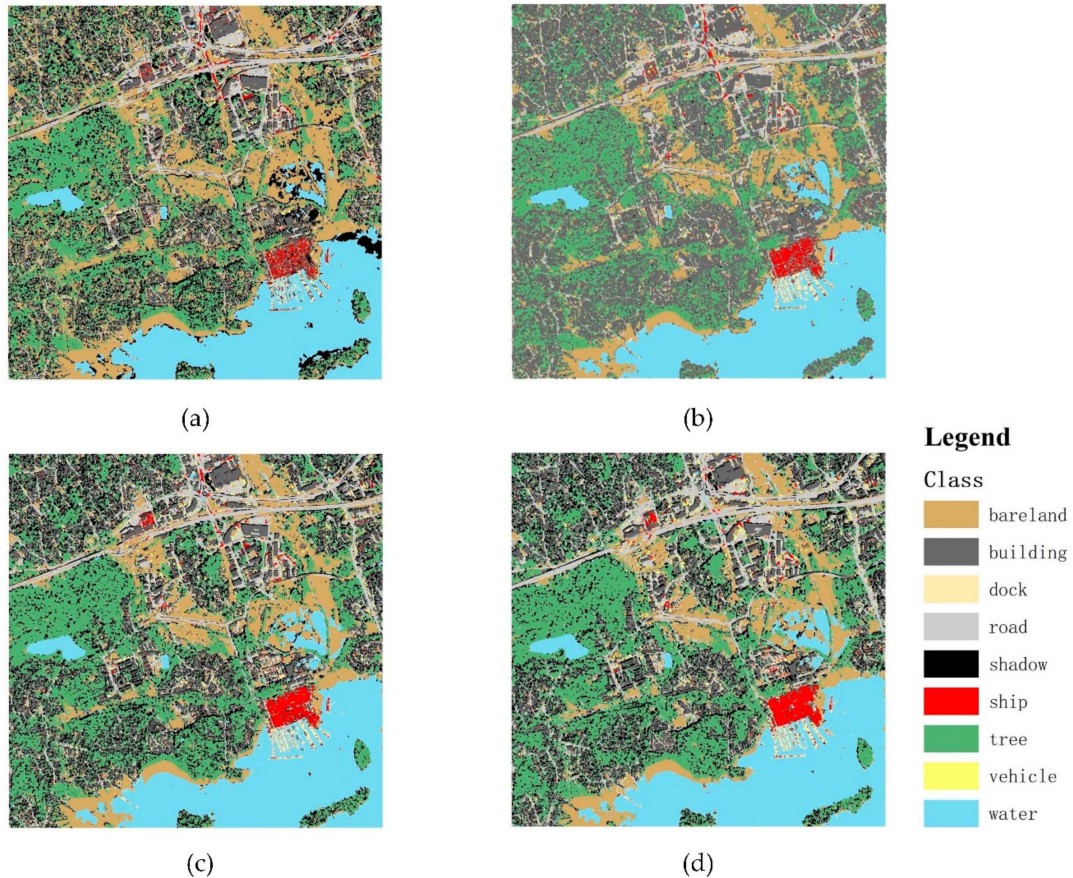

**Figure 15.** The demonstration of four single scale classification results. From (**a–d**) the scale is 15, 25, 35 and 45 respectively.

As shown in Figure 15, the visual effects of four single scale classification results were mainly alike. However, there were still minor differences among them. In Figure 15a (scale 15), some water areas were misclassified into shadows while some shadow areas were misclassified as buildings in Figure 15b (scale 25). In Figure 15c (scale 35) and Figure 15d (scale 45), misclassifications could hardly be seen visually. It should be noted that scale 15 performed best in classifying ships. Moreover, classification results of trees, roads, bare lands and docks were basically similar in all four single scales. The training accuracies of each single scale were respectively 0.876 (scale 15), 0.918 (scale 25), 0.932 (scale 35) and 0.916 (scale 45). Apparently, the training accuracy generally accorded with the classification results shown in Figure 15. However, the final OA and kappa coefficient presented a different result. The OAs of single scale 15 to scale 45 were respectively 0.9016 (scale 15), 0.8719 (scale 25), 0.8856 (scale 35) and 0.8932 (scale 45), while the kappa coefficients were 0.9016 (scale 15), 0.8719 (scale 25), 0.8856 (scale 35) and 0.8932 (scale 45). The finest outcome came from scale 15 while scale 35 owned the highest training accuracy. The cause for such a situation may be the randomness of voters.

The result of multiscale 15–45 which generated the finest classification result among all single scales and their combinations is as demonstrated in Figure 16 along with the original image.

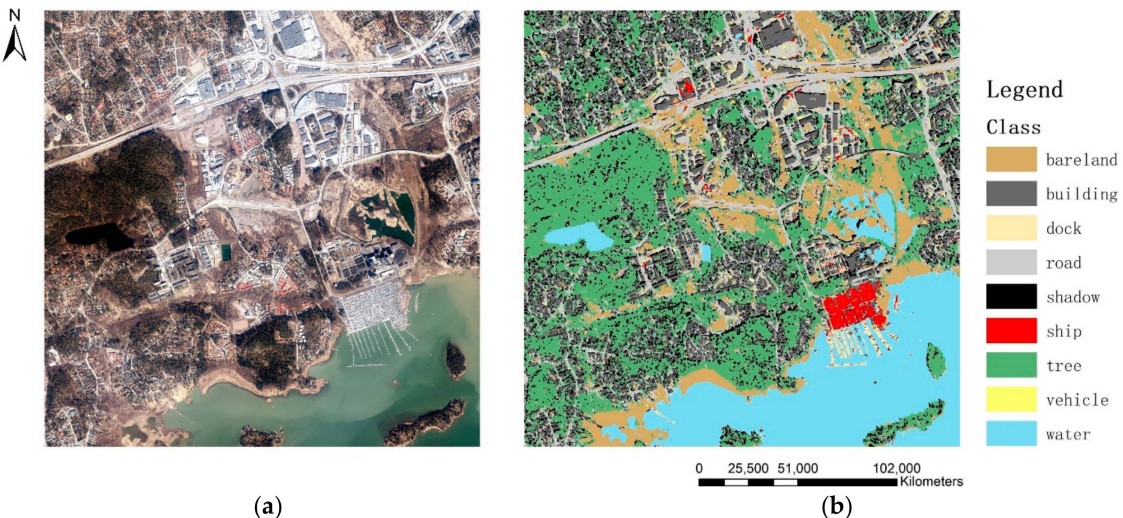

**Figure 16.** The demonstration of experiment area (**a**) and final classification result (**b**).

As Figure 16 shows, the classification results of scale 15–45 was basically identical to the ground truth and remained as the optimal among all fourteen scales (or scale combinations). To be specific, the training accuracy of scale 15–45 was 0.9441 and the OA was 0.9095 which was the highest obtained. In Figure 17, the confusion matrices of the top four classification results are demonstrated.

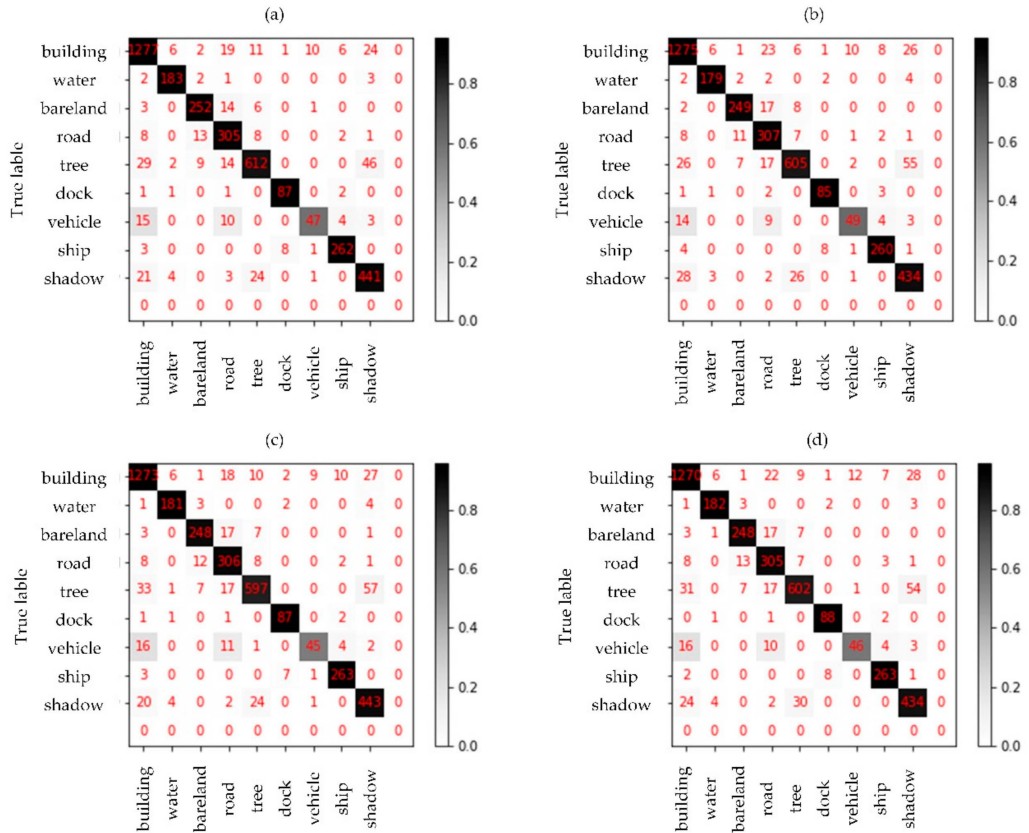

**Figure 17.** The demonstration of confusion matrix. (**a**) Confusion matrix of multiscale 15–45, (**b**) 15–25–45, (**c**) 15–35–45 and (**d**) 25–45.

In Table 6 and Figure 18, the training accuracies, OAs and kappa coefficients of all training scales are demonstrated.

**Table 6.** The classification results of all scales.

| Scale | Training Accuracy | Overall Accuracy |
|---|---|---|
| 15 | 0.876 | 0.9016 |
| 25 | 0.918 | 0.8719 |
| 35 | 0.932 | 0.8856 |
| 45 | 0.916 | 0.8932 |
| 15–25 | 0.9238 | 0.8767 |
| 15–35 | 0.9446 | 0.9 |
| 15–45 | 0.9441 | 0.9095 |
| 25–35 | 0.939 | 0.889 |
| 25–45 | 0.9451 | 0.9021 |
| 35–45 | 0.931 | 0.8971 |
| 15–25–35 | 0.9428 | 0.8969 |
| 15–25–45 | 0.9487 | 0.9034 |
| 15–35–45 | 0.9508 | 0.9034 |
| 25–35–45 | 0.8992 | 0.8992 |

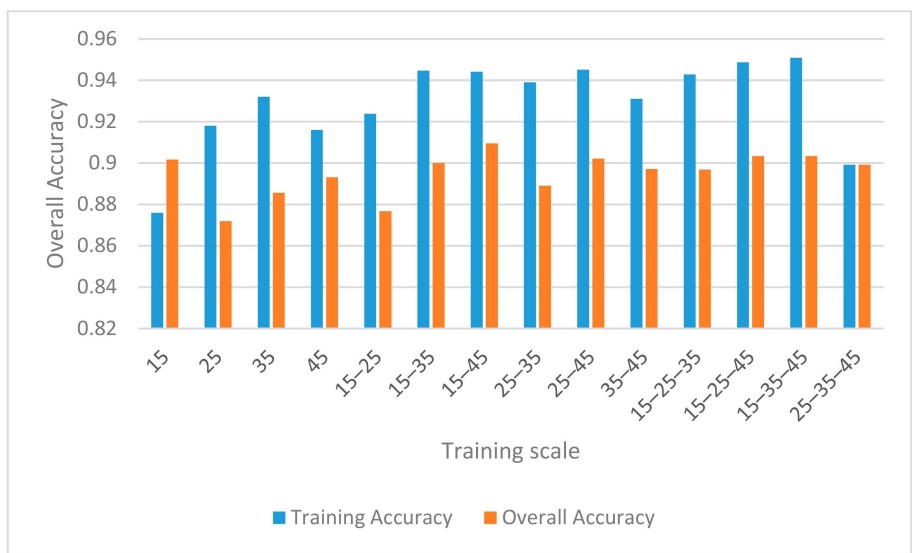

**Figure 18.** Demonstration of classification results.

As Table 6 and Figure 17 indicate, the trend that training accuracies of single scales were lower than contacted multi-scales is apparent, by which overall accuracies abided. Generally, performances of contacted multi-scales were better than that of single scales by a slight margin. Moreover, a VHSRI was input into the tuned network as well and obtained corresponding training accuracies at four single scales: 0.85 (scale 15), 0.9 (scale 25), 0.92 (scale 35) and 0.9 (scale 45).

The final classification OA of nDSM–VHSRI fusion data were as follow: single-scale 15 was 0.814, single-scale 25 was 0.884, single-scale 35 was 0.886, single-scale 45 was 0.89 and the finest result which was obtained in multi-scale 25–35–45 was 0.904. Despite the excellent result generated by nDSM–VHSRI fusion data, StdnDSM fusion data still beat it in OA.

## 4. Discussion

Influences of diverse sampling methods and different parameters in CNN on classification results and computing efficiency will be shown and discussed in this part. Furthermore, superiorities and imperfections of adopted fusion data and stratified multiresolution segmentation will be talked over as well here.

*4.1. Influences of Sampling Strategy and CNN Parameters*

4.1.1. Sampling Strategy

As introduced in Section 2.6.2., this paper adopted the strategy of sampling training and validating points simultaneously. To be specific, samples were randomly divided into training points and validating points after sampling had been done. Such a sampling strategy ensured that training points and validating points abided by the exact same sampling principle. The advantage of this sampling strategy was that no training points would accidentally coincide with validating points and therefore avoided extortionately high overall accuracy. Different sampling strategies were performed and corresponding accuracies of each land-cover categories are compared in Table 7. More specifically, the adopted sampling strategy contained 25,000 training points and 4811 validating points that were sampled at the same time and randomly generated (synchronous sampling), while the contrast sampling strategy consisted of 25,000 training points and 2850 validating points that were asynchronously sampled (asynchronous sampling).

**Table 7.** Accuracy comparison of synchronous and asynchronous sampling strategy.

| Categories | Accuracy by Synchronous Sampling | Accuracy by Asynchronous Sampling |
|---|---|---|
| Bare Land | 90.65% | 92.43% |
| Building | 93.97% | 93.41% |
| Dock | 90.63% | 92.53% |
| Road | 86.4% | 88.92% |
| Shadow | 85.3% | 89.67% |
| Ship | 94.93% | 94.56% |
| Tree | 92.59% | 93.5% |
| Vehicle | 79.66% | 85.79% |
| Water | 93.37% | 95.62% |
| Overall | 90.95% | 91.53% |

Table 7 shows that accuracies of asynchronous sampling strategy in both individual categories and overall samples generally tended to be numerically higher than that of synchronous sampling strategy. However, greater numbers do not necessarily represent better performance. As training samples and validating samples were asynchronously sampled, the possibility that those two types of samples may have coincided and sampling principles may not be exactly the same. If, however, one of those situations occured, the final accuracies may have varied accordingly. In this paper, asynchronous sampling generated validating samples that coincided with or near training points, which consequently increased final accuracies in error. Therefore, the final outcomes of synchronous sampling may be less pretty than that of asynchronous, but the rationality and reliability of synchronous sampling were superior.

4.1.2. CNN Parameters Setting

Alexnet contains numerous parameters that affect training and final classification results. However, only several parameters, namely, batch_size, epoch_number and data_size, are discussed in detail here. While learning_rate and drop_rate were respectively set as 0.01 and 0.5, as they are commonly adopted empiric values.

Batch_size is the number of divided images put into the network. Batch_size determines the speed of training which indirectly points to the efficiency of the network if training accuracies are the same. In our experiments, batch_size hardly impacted training accuracies of the network. Therefore, a larger batch_size here meant a higher training efficiency. However, the setting of batch_size was not the higher the better, as the increase of batch_size theoretically did lower training accuracies. Batch_size

was eventually set as 100, as the training accuracies stabilized at 0.84–0.94, while the training accuracies of batch_size 50, 150 and 200 were unstable and lower.

Epoch_number is the times the workflow of the network runs throughout the training. More specifically, epoch_number determines how many times the CNN will learn the training data and directly decides the value of training accuracy. Generally, training accuracy rises as epoch_number increases till a peak and slightly drops afterwards. Therefore, the final epoch_number was set as 160 after trial and error. Experiments setting epoch_numbers as 50, 100, 150 and 200 were performed as well, yet training results indicated that the maximum training accuracy emerged around 160 epochs.

Data_size is the scale of training or the size of the training sample. In the procedure of the designed experiment, training samples were selected as points, whereas CNN only accepts raster data (small images). Thus, a mutable window centred on sampled training points generated the required data for CNN. As demonstrated in Section 2.6.4, four single sizes (scales) were generated and contacted later. The train and loss curve of the lowest single scale (scale 15), highest single scale (scale 35) and highest multiscale (scale 15–35–45) are demonstrated as follows in Figure 19.

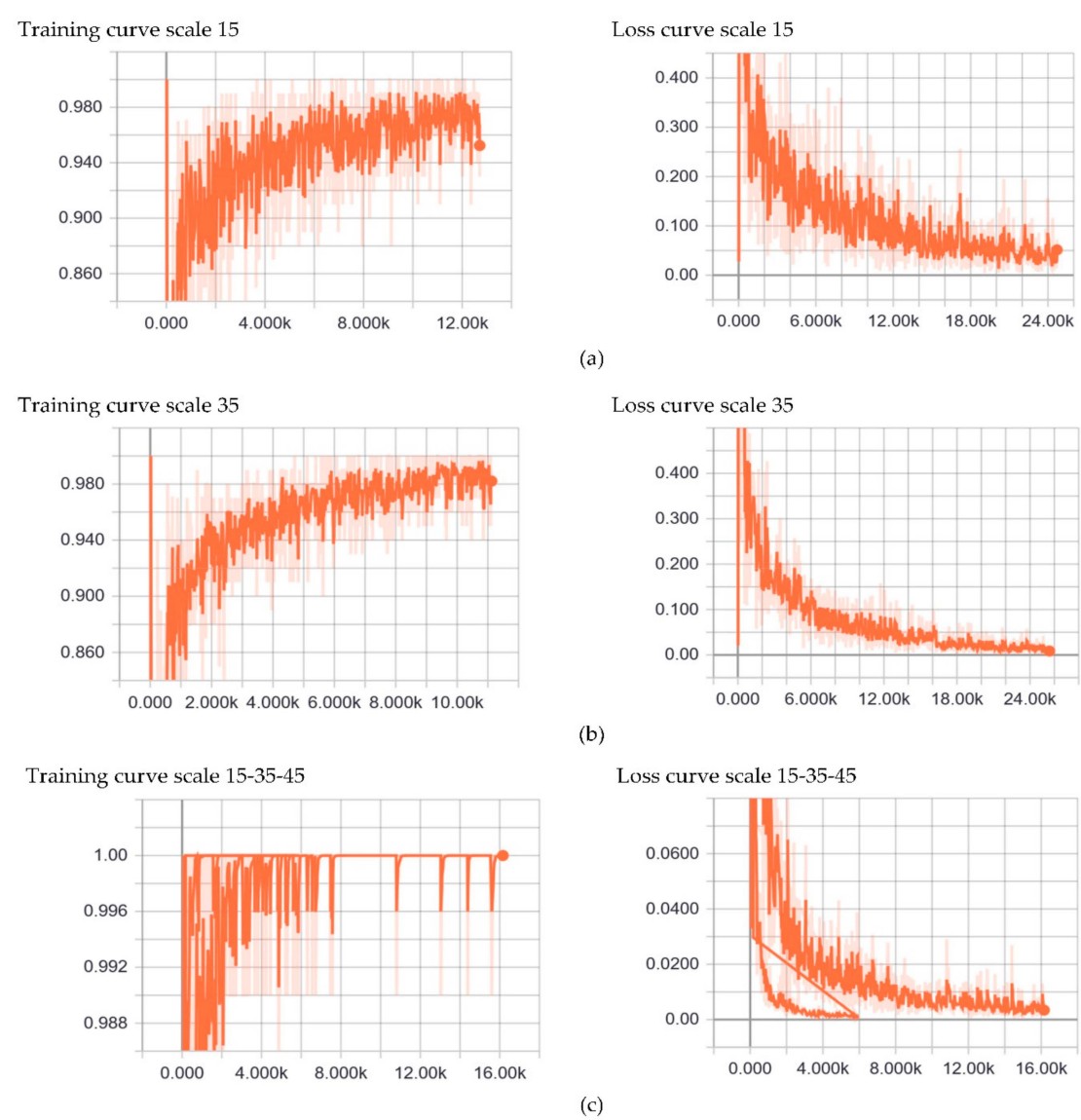

**Figure 19.** The demonstration of train and loss curve. (**a**) The train and loss curve of scale 15, (**b**) scale 35 and (**c**) multiscale 15–35–45.

The training accuracy of scale 15 (Figure 19a) was the lowest of all scales and was 0.876. The highest single scale training accuracy was 0.932 from scale 35 (Figure 19b). Figure 19c shows the highest training accuracy of all scales, 0.9508 from scale 15–35–45. In the light of the training accuracies shown in Table 6 and curves shown in Figure 19, it is apparent that training accuracy increased with the rise of training scale and scale combinations obviously overshadow single scale in accuracy.

### 4.2. Pros and Cons of Point Cloud added Fusion Data and Stratified MRS

As listed in previous sections, superiorities of fusing point clouds (LiDAR data) and VHSRIs (RGB image) by the proposed StdnDSM lie in two main aspects: (1) fusion data contained elevation information which RGB images failed to comprise and (2) StdnDSM evened the ratio of fusion bands and therefore enhanced the amount of information.

The addition of elevation feature encouraged boundary differentiation in both segmentation and training, which were reflected in segmentation quality and training accuracies. The boundary differentiation in segmentation of PC added and non-added data are demonstrated in Figure 20.

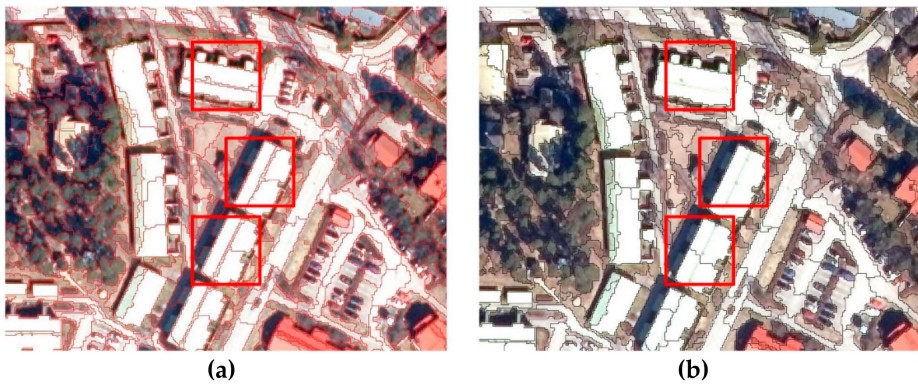

|     |     |
| :-: | :-: |
| (a) | (b) |

**Figure 20.** The comparison of point cloud (PC) added and non-added data in segmentation. Red line (**a**) represents RGB and black line (**b**) is PC added.

As Figure 18 clearly shows that PC added data (StdnDSM fused) surpassed PC non-added data (RGB image) in the segmentation results. Boundaries of buildings and trees where distinct differences in height existed most effectively tell the advantage of PC added data. However, the elevation was not the answer to all problems. Objects like road and bare land were not easily distinguished through StdnDSM and the fusion data was not visual-friendly. Most importantly, the synchrony of point cloud and VHSRIs was hardly guaranteed as such open-sourced data is rare. The data used in this experiment were asynchronous and may be controversial for fusion owing to the inconsistency of moving objects, namely vehicles and ships. However, such objects should not be neglected in classification and most objects, such as buildings, docks, waters and roads, remained basically the same.

Moreover, combining PC and VHSRIs surprisingly mitigated the scale effect in traditional RS image analysis. The possible cause of such a phenomenon was that the addition of elevation defined the boundaries of objects that traditional RGB images failed to specify. However, this explanation is yet to be confirmed and remains a future research subject.

When it comes to stratified MRS, the superiorities of Stratified MRS have been emphasized more than once previously. Above all, the theoretical rationality and computational high efficiency are two unprecedented traits compared to traditional MRS. However, stratified MRS has its own flaws that performing stratified MRS rather complex and cross-platform, while traditional MRS can be easily done on any specialized software. A refined stratified MRS or involving parallel stratified MRS is still a task worth studying in the future.

## 5. Conclusions

Land-cover classification has always been a critical task in remote sensing. Thanks to the rapid developments in RS sensors, massive images of sub-meter resolution are obtained every day, which poses the challenge to automatically and efficiently conduct those data. Therefore, CNN was applied in image classification and proved as one of the finest classifiers. For land cover classification by using images with spatial resolution of meter level or sub-meter level, the designed methodology contained the novel proposed StdnDSM which added elevation information into CNN and adopted stratified multiresolution segmentation and majority voting who both significantly improved the computation efficiency. Conclusions concerning StdnDSM, stratified multiscale segmentation and RMVCNN are drawn as follows.

The proposed StdnDSM was actually an improved nDSM which rendered a smoother fusion of LiDAR data and VHSRIs. The smooth fusion of two types of data balanced the influences of LiDAR data and VHSRIs in segmentation and classification. Compared to preceding fusion data or single-sourced data, the fusion of StdnDSM and VHSRIs was overall theoretically and practically better. Moreover, as mentioned in Section 4.2., the combination of point cloud and VHSRIs alleviated the influences of scale effect in segmentation and classification. Yet, the mechanism behind such a phenomenon is yet to be studied.

Stratified MRS was actually a segmentation strategy which divided an image into several homogenous regions for further segmentation. The stratified MRS was utilized in land-cover classification for the first time here and generated a promising result. The introduction of stratified MRS significantly heightened the segmentation performance in boundaries and efficiency. Thus, adopting the stratification strategy in segmentation was logical and greatly beneficial to object-based classification.

Majority voting strategy is the indispensable part in object-based classification which connects the results of segmentation and CNN. Compared to currently and widely used centre point-oriented strategy which directly renders the category of the centre point to segmented polygon, majority voting avoids misclassifications caused by misclassifying centre points. Therefore, it had better robustness and higher fault-tolerance.

All in all, fusing LiDAR data and VHSRIs is feasible and effective for both image segmentation and land-cover object classification. However, the asynchrony of capture is yet to be addressed. The adopted Alexnet is of high-performance in image classification, nevertheless, new networks, such as graph neural network (GNN), also show great potential and need further exploration. Moreover, the introduction of stratified MRS in the land-cover classification generated promising results and showed impressive efficiency. However, a convincing assessment criteria system for MSR is not yet established and requires further efforts.

**Author Contributions:** Conceptualization, K.Z. and D.M.; methodology, K.Z.; software, K.Z. and X.L.; validation, K.Z. and J.F.; formal analysis, K.Z.; investigation, K.Z.; resources, K.Z.; data curation, K.Z. and J.F.; writing—original draft preparation, K.Z.; writing—review and editing, K.Z. and D.M.; visualization, K.Z.; supervision, M.W.; project administration, D.M.; funding acquisition, M.W.

**Funding:** This research was funded by National Natural Science Foundation of China, grant number 41671369 and 41671341, National Key Research and Development Program, grant number 2017YFB0503600 and ''the Fundamental Research Funds for the Central Universities''.

**Acknowledgments:** Special thanks to the National Land Survey of Finland (NLSF) for providing data. Last but not least, this paper is not the work of a single person but of a great team including members of Lab Ming of China University of Geosciences (Beijing), and some close friends who have helped a lot.

**Conflicts of Interest:** The authors declare no conflict of interest. The funders had no role in the design of the study; in the collection, analyses, or interpretation of data; in the writing of the manuscript, or in the decision to publish the results.

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
