# Peer review of "CNN-Based Land Cover Classification Combining Stratified Segmentation and Fusion of Point Cloud and Very High-Spatial Resolution Remote Sensing Image Data"

_remotesensing, doi:10.3390/rs11172065_

Round 1

Reviewer 1 Report

The introduction of the new criterion StdnDSM derived from DEM/DSM and it's impact in interpretation of the objects included in high res images is the key issue of this paper. 

To my opinion the DSM depicts the highest in elevation points while DTM (digital terrain model) depicts the reference surface (for example ground elevation). Thus in scientific terminology terms DEM should be replaced by DTM.

DEM (digital elevation model) is a confilcting term that in some papers refers to DSM and some other papers to DTM. 

Author Response

Dear reviewer,

We are very excited to have been given the opportunity to revise our manuscript. Herein, we also want to extend our appreciation for your taking the time and effort to provide such insightful comments and guidance. We carefully considered your comments. The revision, based on the review team’s collective input, includes a number of positive changes, and all changes in the manuscript are highlighted by the track changes mode. The response is listed below.

Thanks again.

Best regards,

Keqi Zhou and Dongping Ming

Point 1. The introduction of the new criterion StdnDSM derived from DEM/DSM and it's impact in interpretation of the objects included in high res images is the key issue of this paper. To my opinion the DSM depicts the highest in elevation points while DTM (digital terrain model) depicts the reference surface (for example ground elevation). Thus in scientific terminology terms DEM should be replaced by DTM.DEM (digital elevation model) is a confilcting term that in some papers refers to DSM and some other papers to DTM.

Response 1: Thanks for your kind reminding. We have confirmed and rewritten this part by content-wise, please see Line 23, Line 154-168, Line 188-193, Line 202-213, Line 366 and Line 389.

Reviewer 2 Report

This research presented a new method to improve traditional and Convolutional Neural Network (CNN) Geographic Object-based Image Analysis (GeOBIA) land-cover classification method. The new method involves fusing high spatial resolution images and cloud points. The authors concluded that compared to pixel-based and traditional object-based classification method, the adopted Majority Voting strategy has stronger robustness and avoids misclassifications caused by minor misclassification center points; and indicates that the StdnDSM is the optimal digital elevation model for data fusion with remote sensing images, and the voting strategy based CNN classification is promising in object-based classification. The reviewer believes that the current version of the manuscript is not yet ready for publication; the authors are encouraged to consider the following comments and suggestions and revise the manuscript accordingly.

The authors should consider streamlining the Abstract section. Currently, the Abstract section is very long and it provides a lot of information. The authors should make it more concise. The authors should also consider splitting the Introduction section into two sections, including an Introduction section and a Background (or Related Work) section. The introduction section should focus on introducing the research objectives and research questions, while the Background section should focus on literature review of related work and defining the research gap. The authors should also beef up their literature section. The authors should use more generally accepted terms for this paper. For example, in the remote sensing field, there are four types of resolutions, including spatial resolution, spectral resolution, radiometric resolution, and temporal resolution. In addition, DEM is not a special pattern of DTM. DEM is DTM; these two terms are used interchangeably. Furthermore, in remote sensing, the difference between DSM and DEM (or DTM) is called Digital Height Model (DHM). The authors should use high-spatial resolution to characterize their remote sensing data. The reviewer suggests the authors find a researcher that has a remote sensing background to review the manuscript for using the appropriate terms. The authors mentioned that the LiDAR point clouds were used to process DSM and DEM. What are the attributes of the LiDAR data used for this study? What is their quality level? Who collected the data? What is the ground sampling distance? What is the density of the LiDAR data (points per square meter)? When processing the LiDAR point cloud, what return was used for creating DSM? What return was used for creating DEM? What is the cell assignment type (average, maximum, minimum, IDW, and nearest)? What is the void fill method (linear or nearest neighbor)? The authors did not mention anything about the LiDAR point clouds processing, which prevents other researchers from replicating their research. What is the vertical accuracy of the LiDAR data? Will vertical accuracy impact the outputs? For the architecture of Alexnet, what is the rationale behind of using the different size and dimension? How to make sure these sizes and dimensions are the most effective ones? The conclusions are for very high-spatial resolution remote sensing images. However, the authors did not mentioned what specific spatial resolution this study applies to. Very high-spatial resolution usually has a ground sampling distance of 1 inch or less (centimeter level or even millimeter level). The authors should provide a range of spatial resolution that this research applies. Most of the figures need to be improved. For example, in Figure 1, there are many images are not labeled. In Figure 4, image (d) is not captioned. For figure 15, the reviewer has to zoom in at least 200% to be able to read. If at all possible, please create vector images for readability.

Author Response

Dear reviewer,

We are very excited to have been given the opportunity to revise our manuscript. Herein, we also want to extend our appreciation for your taking the time and effort to provide such insightful comments and guidance. We carefully considered your comments. The revision, based on the review team’s collective input, includes a number of positive changes, and all changes in the manuscript are highlighted by the track changes mode. The response is listed below.

Thanks again.

Best regards,

Keqi Zhou and Dongping Ming

Point 1. The authors should consider streamlining the Abstract section. Currently, the Abstract section is very long and it provides a lot of information. The authors should make it more concise.

Response 1: Thanks for your kind suggestion about the Abstract section. We have carefully streamlined Abstract section, removed unnecessary qualifiers and made the section more concise.

Point 2. The authors should also consider splitting the Introduction section into two sections, including an Introduction section and a Background (or Related Work) section. The introduction section should focus on introducing the research objectives and research questions, while the Background section should focus on literature review of related work and defining the research gap.

Response 2Thank you for your kind suggestion. We agree that the Introduction section is too long and we confirm that it is better to splitting the Introduction section into two sections as you suggested. However, as the editor required, the structure of this paper must be "Introduction-Materials and Methods-Results-Discussion-Conclusions". Splitting the Introduction section comes impossible. However, we have added some topic sentences in each paragraph to strengthen the manuscript structure, please see Line55, 79, 121 and 129.

Point 3. The authors should also beef up their literature section.

Response 3: Thank you for your reminding. As mentioned above, for the limitation of manuscript structure requirement by MDPI, we can’t change the structure of the manuscript, so what we could do is to try to stress the organization of the literature review section by rewrite some sentence and adding some topic sentence in this section. Please see Line 64-78, 99-102, 134, 137; Line55, 79, 121 and 129.

Point 4. The authors should use more generally accepted terms for this paper. For example, in the remote sensing field, there are four types of resolutions, including spatial resolution, spectral resolution, radiometric resolution, and temporal resolution.

Response 4: Apologize for unclear terms utilized. The term VHRI (Very High-Resolution Image) was used in some cited papers and well accepted among certain groups of researchers. However, the unclear reference do exist in VHRI. Thank you for your kind reminding. In this revision, we have replaced VHRI with VHSRI (Very High-Spatial Resolution Image) on content-wise.

Point 5. In addition, DEM is not a special pattern of DTM. DEM is DTM; these two terms are used interchangeably. Furthermore, in remote sensing, the difference between DSM and DEM (or DTM) is called Digital Height Model (DHM).

Response 5: Thanks for your kind reminding. We have confirmed and replaced DEM with DTM by content-wise. For DHM, as nDSM is also accepted widely in remote sensing, we have noted that DHM equals to nDSM, please see Line 155 and Line 156.

Point 6. The authors should use high-spatial resolution to characterize their remote sensing data. The reviewer suggests the authors find a researcher that has a remote sensing background to review the manuscript for using the appropriate terms.

Response 6: Thank you for your suggestion. We have asked a professor from our university to help us with the terms used in this paper.

Point 7. The authors mentioned that the LiDAR point clouds were used to process DSM and DEM. What are the attributes of the LiDAR data used for this study? What is their quality level? Who collected the data? What is the ground sampling distance? What is the density of the LiDAR data (points per square meter)? When processing the LiDAR point cloud, what return was used for creating DSM? What return was used for creating DEM? What is the cell assignment type (average, maximum, minimum, IDW, and nearest)? What is the void fill method (linear or nearest neighbor)?

Response 7: Sorry for unclear introduction about the LiDAR data. The attributes of the collected LiDAR data used in this study were well introduced in section 2.6.1., please see Line 378-Line 389. As for the return number used to generate DTM and DSM, the description is yet unclear. In this revision, we have added the details that the process was performed on software LiDAR360 mentioned in Line 392. For cell assignment type and the void fill method, the engineer from LiDAR360 has confirmed with us that the cell assignment type is IDW and the void fill method is nearest neighbor. We have supplemented these information in the revised manuscript, please see Line 392-Line 394.

Point 8. What is the vertical accuracy of the LiDAR data? Will vertical accuracy impact the outputs?

Response 8: Sorry for unclear explanation. As a fact, the vertical accuracy of the LiDAR data was also introduced in section 2.6.1., please see Line 384. In my opinion, vertical accuracy impact the outputs little in our work (The elevation precision is 0.15m while the average point density(resolution) is 43pis/m2), however the accurate influence of vertical accuracy on outputs is currently not probed and remain as our future study task.

Point 9. For the architecture of Alexnet, what is the rationale behind of using the different size and dimension? How to make sure these sizes and dimensions are the most effective ones?

Response 9: We chose those scales and set those parameters is actually out of empirical analysis and trial and error. Dozens of experiments were conducted to gain those parameters in this work. Though we have originally proposed some spatial statistical method for remote sensing scale processing, such as for segmentation scale selection or for adaptive scale selection in CNN classification (more details please see our previous publications listed below), it is almost impossible to absolutely affirm the most effective ones for all different data because the complexity of spatial data, especially when the spatial extent of the image data is very broad and the landscape structure is very complex.

Yangyang ChenDongping Ming*Xianwei Lv.Superpixel based land cover classification of VHR satellite image combining multi-scale CNN and scale parameter estimation. Earth Science Informatics,2019, DOI: 10.1007/s12145-019-00383-2. Dongping Ming*,Xian Zhang,Min Wang,Wen Zhou. Cropland Extraction Based on OBIA and Adaptive Scale Pre-estimation, Photogrammetric Engineering & Remote Sensing, 2016, 82(8): 635–644. doi: 10.14358/PERS.82.8.635. Dongping Ming*,Jonathan Li,Junyi Wang,Min Zhang. Scale parameter selection by spatial statistics for GeOBIA: Using mean-shift based multi-scale segmentation as an example. ISPRS Journal of Photogrammetry and Remote Sensing, 2015, 8, 106:28-41. DOI: 10.1016/j.isprsjprs.2015.04.010. Ming*, T. Ci, H. Cai, L. Li, C. Qiao, J. Du. Semivariogram based spatial bandwidth selection for remote sensing image segmentation with mean-shift algorithm, IEEE Geoscience and Remote Sensing Letters,20129(5),813-817.

Point 10. The conclusions are for very high-spatial resolution remote sensing images. However, the authors did not mentioned what specific spatial resolution this study applies to. Very high-spatial resolution usually has a ground sampling distance of 1 inch or less (centimeter level or even millimeter level). The authors should provide a range of spatial resolution that this research applies.

Response 10: Sorry for unclear explanation. In the revision, we have stressed that the proposed method is suitable for images with spatial resolution of meter level or sub-meter level, please see Line 687-688.

Point 11. Most of the figures need to be improved. For example, in Figure 1, there are many images are not labeled. In Figure 4, image (d) is not captioned. For figure 15, the reviewer has to zoom in at least 200% to be able to read. If at all possible, please create vector images for readability.

Response 11: Thank you for your kind reminding and suggestion. The figures mentioned by you have been remapped. Figure 1 has been revised and all images are now labeled, please see Line 150. Figure 4 has been revised and image (d) is now captioned as well, please see Line 207. For Figure 15, we couldn’t manage to create vector image but we did improve the quality of the image and now it is perfectly readable. Please see Line 550.

Reviewer 3 Report

This manuscript (remote-sensing-574938) proposed a multi-source image classification based on CNN algorithm using Very High-Resolution images and point cloud. The authors converted the point cloud to raster format which is compatible with deep learning approaches. The manuscript needs English proof-reading because there are several grammatical issues and typography errors in there. I also do have some concerns as follows:

Abstract: The abstract in the current version is too long. The abstract should be short enough for readers to scan quickly and long enough to give them enough information to decide to read the article. The abstract should be rewritten another time since, in the current version, it is more similar to an introduction. In the abstract you need to indicate the type of information found in the paper; explains the purpose, objective, and methods of the paper as well as your contribution and show the improvements that were obtained compared to other methods. In the introduction, authors should add some explanation about two types of CNN models that are commonly used in remote sensing applications: patch-based vs Fully convolutional. Add information about training time and the advantages of each method. They can use and cite these papers for readers to address it: Convolutional neural networks for large-scale remote-sensing image classification. IEEE TGRS, 2016. Very deep convolutional neural networks for complex land cover mapping using multispectral remote sensing imagery, Remote Sensing journal, 2018. Authors should add training accuracy and loss curve in section 4.1.2. Add scale bar in Figure 16. Add confusion matrix for final classification result. I suggest to authors that compared the result with another machine learning approach to illustrate the improvement of the proposed method. In conclusion, the authors should add one or two sentences about the interesting results obtained in this manuscript.

Author Response

Dear reviewer,

We are very excited to have been given the opportunity to revise our manuscript. Herein, we also want to extend our appreciation for your taking the time and effort to provide such insightful comments and guidance. We carefully considered your comments. The revision, based on the review team’s collective input, includes a number of positive changes, and all changes in the manuscript are highlighted by the track changes mode. The response is listed below.

Thanks again.

Best regards,

Keqi Zhou and Dongping Ming

Point 1. The abstract in the current version is too long. The abstract should be short enough for readers to scan quickly and long enough to give them enough information to decide to read the article. The abstract should be rewritten another time since, in the current version, it is more similar to an introduction. In the abstract you need to indicate the type of information found in the paper; explains the purpose, objective, and methods of the paper as well as your contribution and show the improvements that were obtained compared to other methods.

Response 1: Thanks for your kind suggestion about the Abstract section. We have carefully streamlined Abstract section, removed unnecessary qualifiers and made the section more concise.

Point 2. In the introduction, authors should add some explanation about two types of CNN models that are commonly used in remote sensing applications: patch-based vs Fully convolutional. Add information about training time and the advantages of each method. They can use and cite these papers for readers to address it: Convolutional neural networks for large-scale remote-sensing image classification. IEEE TGRS, 2016. Very deep convolutional neural networks for complex land cover mapping using multispectral remote sensing imagery, Remote Sensing journal, 2018.

Response 2Thank you for your kind suggestion. Papers recommended are very help to our understanding of CNN. Comparison between patch-based CNN and FCN has been added in the Introduction, please see Line 100-102.

Point 3. Authors should add training accuracy and loss curve in section 4.1.2. Add scale bar in Figure 16.

Response 3: Thank you for your suggestion. We have added the training accuracy and loss curve obtained by Tensorboard in Figure 19, please see Line 645-647. We have also added a scale bar in Figure 16, please see line 569.

Point 4. Add confusion matrix for final classification result.

Response 4: Thank you for your suggestion. We have added the confusion matrix, please see in Line 574.

Point 5. I suggest to authors that compared the result with another machine learning approach to illustrate the improvement of the proposed method.

Response 5: Thank you for your suggestion. As a fact, many researches have proved that Alexnet is superior to Random forest and SVM methods. So, we mainly pay our effort in comparing the adopted methodology and other DL methods including Graph Neural Network (GNN), which is our current study task. We do hope we could achieve more excited research results.

Point 6. In conclusion, the authors should add one or two sentences about the interesting results obtained in this manuscript.

Response 6: We have added some content concerning our conclusion on scale effect which is also a subject to be thoroughly studied in the near future. Please see Line 687-688 and Line 697-699.

Round 2

Reviewer 2 Report

The authors have addressed all my comments. 

Author Response

Response to Reviewer 3

Dear reviewer,

Thank you for your noble work and kind suggestions.

Thanks again.

Best regards,

Keqi Zhou and Dongping Ming

Reviewer 3 Report

Please find my comments in the attached file.

Author Response

Response to Reviewer 3

Dear reviewer,

We are very excited to have been given the opportunity to revise our manuscript. Herein, we also want to extend our appreciation for your taking the time and effort to provide such insightful comments and guidance. We carefully considered your comments. The revision, based on the review team’s collective input, includes a number of positive changes, and all changes in the manuscript are highlighted by the track changes mode. The response is listed below.

Thanks again.

Best regards,

Keqi Zhou and Dongping Ming

Point 1. There is a problem in citation since there several [] without any number on that and some of them were not properly cited. Author should fix the citation problem in this step.

Response 1: Thanks for your kind reminding. We have carefully corrected all concerning errors throughout the manuscript.

Point 2. According to reviewer 2 comments in the previews version of the manuscript about several studied previously compared Alexnet to RF and SVM. At least the authors should provide a couple of those papers in remote sensing applications and refer readers to them such as:

convolutional neural network for complex wetland classification using optical remote

sensing imagery. IEEE Journal of Selected Topics in Applied Earth Observations and

Remote Sensing, 11(9), pp.3030-3039.

Response 2Thank you for your kind suggestion. Indeed, citations concerning this point lacked and we have added recommended paper, please see Line 83.